**PLOS** COMPUTATIONAL BIOLOGY

# Integrative analysis of ATAC-seq and RNA-seq for cells infected by human T-cell leukemia virus type 1

Azusa Tanaka[1]*, Yasuhiro Ishitsuka[2]*, Hiroki Ohta[3]*, Norihiro Takenouchi[4], Masanori Nakagawa[5], Ki-Ryang Koh[6], Chiho Onishi[7], Hiromitsu Tanaka[8,9], Akihiro Fujimoto[1], Jun-ichirou Yasunaga[10], Masao Matsuoka[10]

**1** Department of Human Genetics, Graduate School of Medicine, The University of Tokyo, Tokyo, Japan, **2** Institute of Mathematics for Industry, Kyushu University, Fukuoka, Japan, **3** Department of Human Sciences, Obihiro University of Agriculture and Veterinary Medicine, Hokkaido, Japan, **4** Department of Microbiology, Kansai Medical University, Osaka, Japan, **5** Department of Neurology, Kyoto Prefectural University of Medicine, Kyoto, Japan, **6** Department of Hematology, Osaka General Hospital of West Japan Railway Company, Osaka, Japan, **7** Laboratory of Ultrastructural Virology, Institute for Life and Medical Sciences, Kyoto University, Kyoto, Japan, **8** Department of Biophysics, Graduate school of Science, Kyoto University, Kyoto, Japan, **9** Department of Developmental Biology, Graduate School of Medicine, Chiba University, Chiba, Japan, **10** Department of Hematology, Rheumatology and Infectious Disease, Faculty of Life Sciences, Kumamoto University, Kumamoto, Japan

\* tanaka.azusa.c10@kyoto-u.jp (AT); ishitsuka.yasuhiro.751@m.kyushu-u.ac.jp (YI); hirokiohta@obihiro.ac.jp (HO)

## Abstract

Human T-cell leukemia virus type 1 (HTLV-1) causes adult T-cell leukemia (ATL) and HTLV-1-associated myelopathy (HAM) after a long latent period in a fraction of infected individuals. These HTLV-1-infected cells typically have phenotypes similar to that of CD4+T cells, but the cell status is not well understood. To extract the inherent information of HTLV-1-infected CD4+ cells, we integratively analyzed the ATAC-seq and RNA-seq data of the infected cells. Compared to CD4+T cells from healthy donors, we found anomalous chromatin accessibility in HTLV-1infected CD4+ cells derived from ATL cases in terms of location and sample-to-sample fluctuations in open chromatin regions. Further, by focusing on systematically selected genes near the open chromatin regions, we quantified the difference between the infected CD4+ cells in ATL cases and healthy CD4+T cells in terms of the correlation between the chromatin structures and the gene expressions. Based on a further analysis of chromatin accessibility, we detected *TLL1* (Tolloid Like 1) as one of the key genes that exhibit unique gene expressions in ATL cases. A luciferase assay indicated that TLL1 has an isoform-dependent regulatory effect on TGF-*β*. Overall, this study provides results about the status of HTLV-1-infected cells, which are qualitatively consistent across the different scales of chromatin accessibility, transcription, and immunophenotype.

**Data Availability Statement:** All ATAC-seq and RNA-seq data needed to reproduce this study have been deposited at the DNA Data Bank of Japan (DDBJ) under accession numbers PRJDB18166 (https://ddbj.nig.ac.jp/search/entry/bioproject/

PRJDB18166) and PRJDB18259 (https://ddbj.nig.ac.jp/search/entry/bioproject/PRJDB18259). In addition to the methods to reproduce all the figures, which we have already described in Materials and Methods, the numerical codes and data to reproduce Figs 1, 2, 4, 5, 7, 8, 9, and 11 are also available from the website with the link of https://osf.io/6gtuy/. The algorithm to reproduce Table 2 is available from the website with the link of https://github.com/tanakanishi/findclosest.

**Funding:** This work was supported by the Research Program on Emerging and Re-emerging Infectious Diseases (21fk0108088h0003 to J.Y. and M.M.) from the Japan Agency for Medical Research and Development (AMED), grant from the Naito Foundation (to A.T.), and KAKENHI (23H02936 to J.Y.; 19H03689 to M.M.; 21K06393 to H.T.; JP19K16740 and JP18J40119 to A.T.) from the JSPS. This study was also supported in part by the JSPS Core-to-Core Program A, Advanced Research Networks (to J.Y. and M.M). The funders had no role in study design, data collection and analysis, decision to publish, or preparation of the manuscript.

**Competing interests:** The authors have declared that no competing interests exist.

## Author summary

Human T-cell leukemia virus type 1 (HTLV-1) causes adult T-cell leukemia (ATL) and HTLV-1-associated myelopathy (HAM) after a long latent period in a fraction of infected individuals. These HTLV-1-infected cells typically have phenotypes similar to that of CD4$^+$T cells, but the cell status is not well understood. To extract the inherent information of HTLV-1-infected CD4$^+$ cells, we integratively analyzed the ATAC-seq and RNA-seq data of the infected cells. Compared to CD4$^+$T cells from healthy donors, we found anomalous properties in terms of chromatin accessibility and its relationship with the gene expressions in HTLV-1 infected CD4$^+$ cells derived from ATL cases. Based on a further analysis of chromatin accessibility, we detected *TLL1* (Tolloid Like 1) as one of the key genes that exhibit unique gene expressions in ATL cases. Indeed, our experiments indicated that TLL1 has an isoform-dependent regulatory effect on TGF-$\beta$. Overall, this study provides results about the status of HTLV-1-infected cells, which are qualitatively consistent across the different scales of chromatin accessibility, transcription, and immunophenotype.

## Introduction

It has been statistically estimated that there are more than 300, 000 types of mammalian host viruses [1]. Among the many viruses that have been discovered, only a few have been reported to cause cancers, such as the DNA virus human papillomavirus (HPV) and the RNA virus hepatitis C virus (HCV) [2]. Human T-cell leukemia virus (HTLV-1) is an oncogenic retrovirus, which is estimated to infect approximately 10 million people worldwide [3].

Adult T-cell leukemia (ATL) and HTLV-1-associated myelopathy (HAM) are both associated with prior infection with HTLV-1. However, these two diseases have different clinical and pathological presentations [4, 5]. The genes encoded by HTLV-1, such as HBZ (HTLV-1 basic leucine zipper factor) and *Tax*, have been reported to affect important signaling pathways involved in cell proliferation, apoptosis, and infectivity [6]. In particular, HBZ is maintained in all ATL cases and functions as both a protein and RNA [7–10]. Recent studies have elucidated that in ATL cells, genomic mutations are highly enriched in T cell-related pathways, such as NF-$\kappa$B, and typically activate the pathways [11, 12]. Furthermore, it has been frequently observed in ATL cases that the aberrant expression of programmed cell death 1-ligand 1 (PD-L1) is caused by disruption of the PD-L1 3'-untranslated region (UTR) [13].

Several questions about these diseases at the genomic scale remain, including how the chromatin structure of ATL cells differs from that of CD4$^+$T cells derived from healthy donors and how this difference influences transcription and translation to finally cause symptoms. In general, cellular phenotypes are largely affected by gene expressions that are strongly correlated with the epigenetic mechanisms occurring in chromatin. To understand the epigenetic mechanisms, it is important to understand how human DNA is packed and chemically modified in the nucleus, which can be quantified by measuring chromatin accessibility.

In this paper, we mainly study the relationship between chromatin accessibility and transcription in HTLV-1-infected cells at the whole genome level using Assay for Transposase-Accessible Chromatin using sequencing (ATAC-seq) [14] and RNA sequencing (RNA-seq) data, based on our previously developed algorithm of systematic clustering of chromatin structures [15]. We performed a comparative analysis of HTLV-1-infected CD4$^+$ cells from ATL cases, HAM cases, and CD4$^+$T cells from healthy donors (healthy CD4$^+$T cells).

Our analysis shows that the infected CD4+ cells derived from ATL cases have anomalous properties in terms of the locations and sample-to-sample fluctuations of open chromatin regions compared with healthy CD4+T cells. We analyzed the correlation between chromatin accessibility and gene expression and found that the infected CD4+ cells in ATL cases show unique properties in the correlation compared to that of healthy CD4+T cells. We also found a relationship between chromatin accessibility and immunophenotype to suggest that some ATL cases are close to several types of myeloid cells.

Finally, we detected *TLL1* (Tolloid Like 1) as one of a few genes having anomalous expressions in ATL cases. A luciferase assay found that TLL1 isoforms, depending on the types of its isoforms, regulate differently the maturation of TGF-β (transforming growth factor β), which is known to play important roles in cancer progression.

## Results

### Chromatin accessibility: Whole view of the genome

The landscape of chromatin accessibility provides useful information for understanding the mechanisms that govern cell-type-specific gene expressions. Preliminarily, we overview the chromatin accessibility characterized by the ATAC-seq of healthy CD4+T cells, ATL cells, and HAM cells.

To obtain the chromatin accessibility landscape, we performed ATAC-seq on HTLV-1-infected CD4+ cells obtained from the peripheral blood of 29 ATL and 6 HAM cases. In this paper, HTLV-1-infected CD4+ cells in ATL and HAM cases operationally mean the cell fraction showing both CD4 and CADM1 positive, which in ATL are thought to be consisted mainly of leukemic cells. All samples selected for the ATAC-seq library preparation were at least 98% HTLV-1-infected cells. The ATAC-seq libraries were sequenced with an average of 44 million reads, resulting in a dataset comprising of 1.3 billion and 556 million sequenced reads for ATL and HAM, respectively. The data quality was high in all cases, with mitochondrial read rates of 7.5% for ATL and 7.3% for HAM. For a comparison, we used ATAC-seq datasets of CD4+T cell samples from 5 healthy donors, which were downloaded from GEO accession GSE74912 [16].

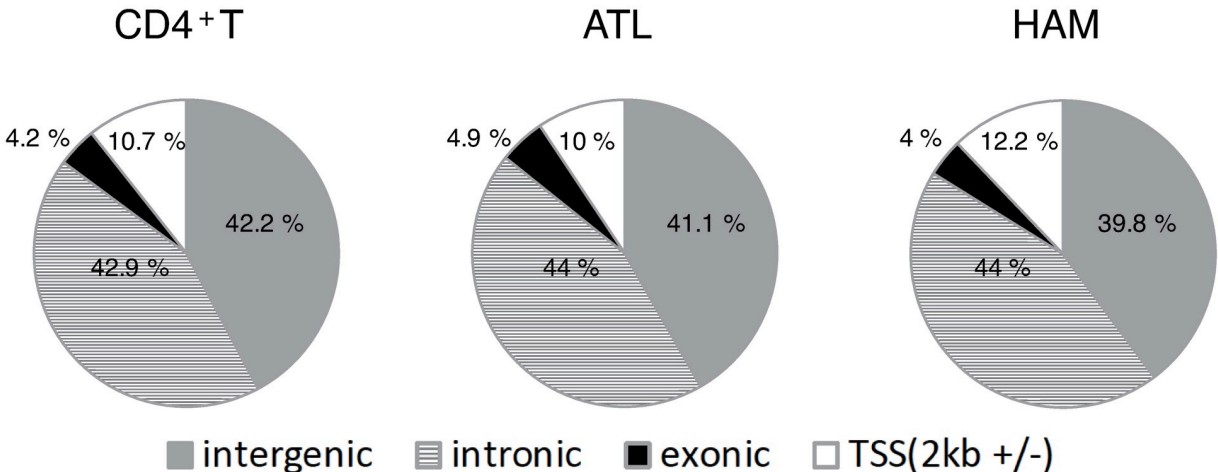

**Fig 1. The genetic (intergenic/intronic/exonic) annotation of ATAC-seq peaks, which quantify open chromatin regions.** TSS(2kb + /−) corresponds to −2000 to 2000 base pairs from a transcription start site.

To identify genome-wide accessible chromatin regions, for each of the three groups of cell types, we concatenated ATAC-seq reads for the different samples, where the sample number was 29 for ATL, 6 for HAM, and 5 for healthy CD4$^+$T cells. As explained in Materials and Methods, we randomly selected 100 million reads from the concatenated data of each group. We used the MACS2 algorithm to select the locations of peaks to quantify the open chromatin regions from the ATAC-seq datasets [17], finding a total of 178811, 89972, and 131609 peaks in ATL, HAM, and healthy CD4$^+$T cells, respectively.

The ENCODE consortium reported that 10% of peaks are localized in near transcription start sites (TSSs), whereas the remaining 90% of peaks are mapped nearly equally to intronic and intergenic regions [18]. Consistent with these data, as shown in Fig 1, about 10% of the ATAC-seq peaks are overlapped with the TSSs and their surrounding regions, whereas the majority of ATAC-seq peaks (about 85%) of healthy CD4$^+$T cells, ATL cells, and HAM cells reside in intergenic or intronic regions.

To determine the functional roles of the peaks in HTLV-1-infected cells and healthy CD4$^+$T cells, we computed the overlapping ratio of these regions with specific genomic features, such as active TSS, enhancers, heterochromatin, etc. To assign a genomic feature to all genomic positions, we assumed a chromHMM 15-state model obtained from (https://egg2. wustl.edu/roadmap/web_portal/chr_state_learning.html). We used the data of E043 for healthy CD4$^+$T cells and E037 for HTLV-1-infected CD4$^+$ cells. Note that E037 is the model of CD4$^+$ memory T cells because a majority of ATL cases has been reported to show CD45RO$^+$, which is consistent with CD4$^+$ memory T cells [19].

As shown in Fig 2 and Table 1, for HTLV-1-infected CD4$^+$ cells, the number of peaks compared with healthy CD4$^+$T cells was proportionally less in categories related to Enhancer (6, 7, 11, 12) and Heterochromatin (9), but it was higher for the category of Quiescent/Low (15) only for ATL cases.

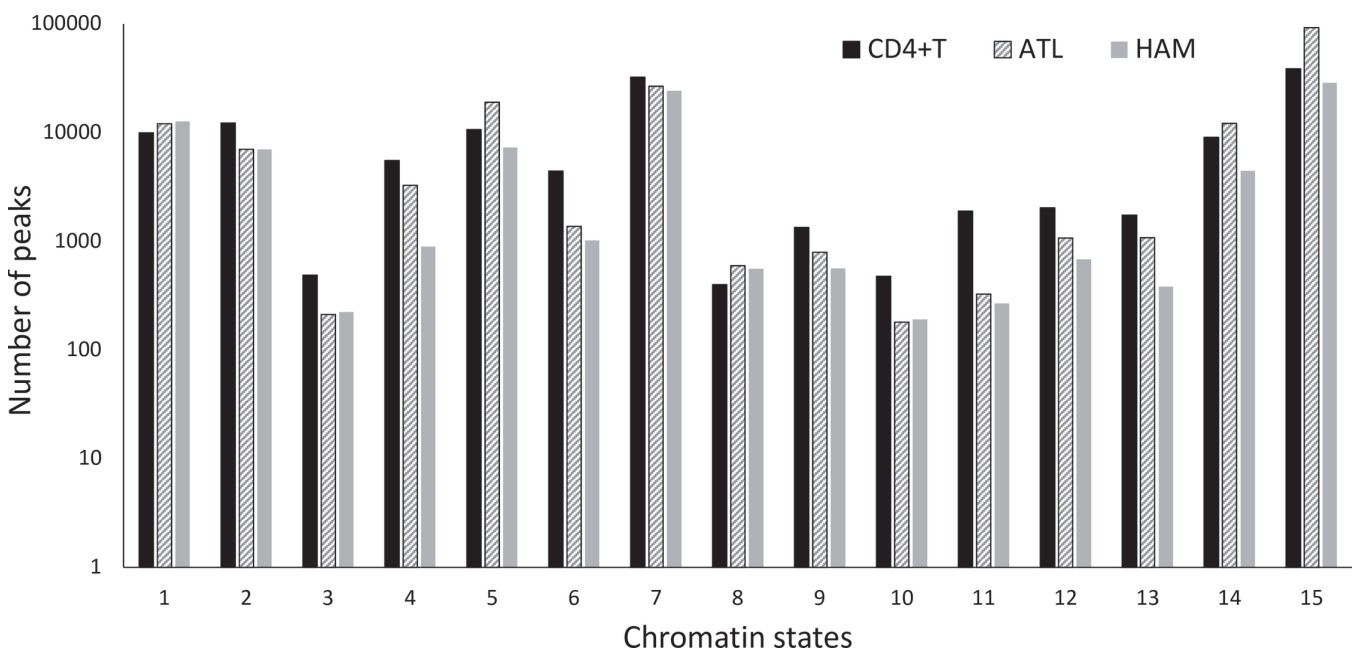

**Fig 2. The number of ATAC-seq peaks (vertical axis) vs. indices (horizontal axis) classified by each functional annotation from 1 to 15, as shown in Table 1.**

**Table 1. The number of ATAC-seq peaks classified into each functional annotation for healthy CD4$^+$T cells, ATL cells, and HAM cells.** $N_{CD4^+T}$, $N_{ATL}$, and $N_{HAM}$: peak number of healthy CD4$^+$T, ATL, and HAM, respectively. Note that each peak quantifies an open chromatin region.

| No. | Description | $N_{CD4^+T}$ | $N_{ATL}$ | $N_{HAM}$ | $\dfrac{N_{ATL}}{N_{CD4^+T}}$ | $\dfrac{N_{HAM}}{N_{CD4^+T}}$ |
|---|---|---|---|---|---|---|
| 1 | Active TSS | 9939 | 12084 | 12691 | 1.22 | 1.28 |
| 2 | Flanking Active TSS | 12202 | 7015 | 6983 | 0.575 | 0.572 |
| 3 | Transcr. at gene 5' and 3' | 489 | 213 | 224 | 0.436 | 0.458 |
| 4 | Strong transcription | 5525 | 3278 | 893 | 0.593 | 0.162 |
| 5 | Weak transcription | 10659 | 18946 | 7269 | 1.78 | 0.682 |
| 6 | Genic enhancers | 4431 | 1374 | 1017 | 0.310 | 0.230 |
| 7 | Enhancers | 32260 | 26597 | 24283 | 0.824 | 0.753 |
| 8 | ZNF genes & repeats | 401 | 595 | 559 | 1.48 | 1.39 |
| 9 | Heterochromatin | 1340 | 794 | 565 | 0.593 | 0.422 |
| 10 | Bivalent/Poised TSS | 478 | 181 | 192 | 0.379 | 0.402 |
| 11 | Flanking Bivalent TSS/Enh | 1894 | 326 | 268 | 0.172 | 0.141 |
| 12 | Bivalent Enhancer | 2031 | 1074 | 682 | 0.529 | 0.336 |
| 13 | Repressed PolyComb | 1746 | 1077 | 382 | 0.617 | 0.219 |
| 14 | Weak Repressed PolyComb | 9086 | 12178 | 4445 | 1.34 | 0.489 |
| 15 | Quiescent/Low | 38566 | 92307 | 28746 | 2.39 | 0.745 |

This observation suggests that compared with healthy CD4$^+$T cells, distinct enhancer mechanisms in HTLV1-infected cases are correlated with the distinct chromatin structures.

## Increased chromatin accessibility around transcriptional start sites (TSSs) in ATL

To determine how the chromatin structures observed in the HTLV1-infected cases are statistically characterized depending on the positions in the genome, we examined the positions of the reads from the ATAC-seq data.

We plotted a histogram $\tilde{\rho}_v(z)$ of the reads as a function of their positions $z$ relative to TSSs for cell type $v$, where $v$ is an ATL, HAM, or healthy CD4$^+$T cell type. Note that the positions of the TSSs and coding regions of all genes were obtained from the human genome (hg19). For technical details of the histogram, see Materials and methods. As shown in Fig 3a, the tail parts of the histogram for ATL cases take higher values compared with HAM and healthy CD4$^+$T cells, whereas the latter two cases showed more similar forms as a whole.

To elucidate the statistics of the chromatin structures around the TSSs, we also focused on the fragments, which can be reconstructed by the reads data; both ends of a fragment correspond to a pair of two reads. Specifically, we investigated the position-dependent accumulation of the fragments, which can be an estimate of nucleosome positioning. We plotted a heat map $F_v^{\Delta,\xi}(z,\ell)$ in which the mid-point of each fragment relative to TSSs is placed on the horizontal axis as $z$ and the length of each fragment is placed on the vertical axis as $\ell$ [20]. For technical details of the heat map, see section Materials and methods.

As shown in Fig 3b, healthy CD4$^+$T cells and HTLV-1-infected CD4$^+$ cells from HAM samples show a pattern of enriched nucleosome-free fragments ($\ell < 100$ bp) and mono-nucleosome fragments ($\ell = 180 \sim 247$ bp) surrounding the TSSs, where the thresholds for the length of a fragment are 100 bp, 180 bp, and 247 bp based on a previous study [21]. ATL samples showed less enrichment of nucleosome-free and mono-nucleosome fragments.

(a)

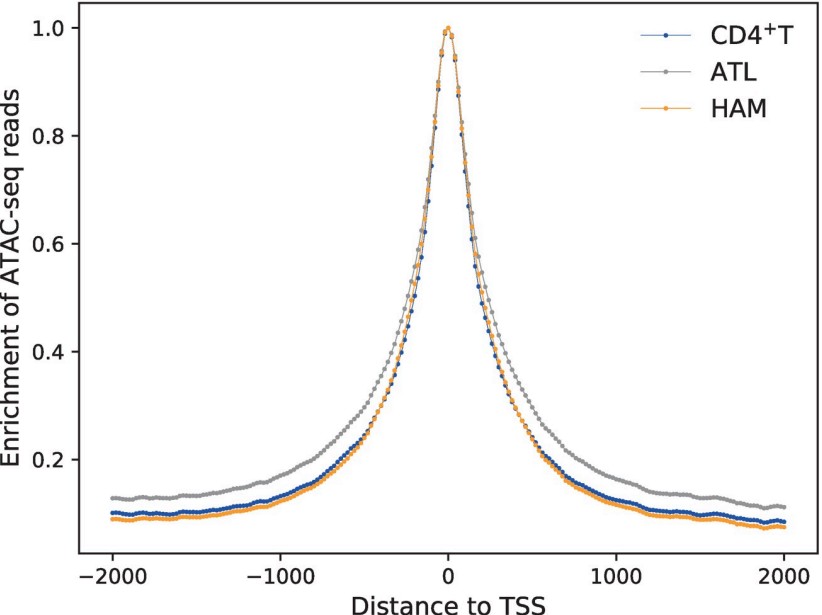

(b)

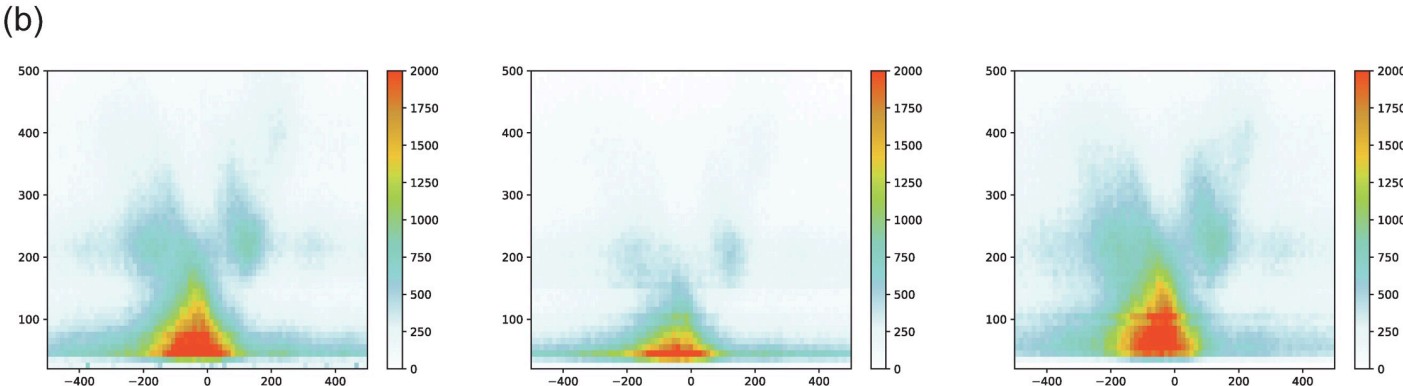

**Fig 3.** (a) The normalized number $\tilde{\rho}_\nu(z)$ of reads at position $z$ from the TSS for type $\nu$ as defined in (4). Note that position $z$ takes the value $\{-2000 + 20k \mid 0 \leq k \leq 200\}$, where $k$ is an integer. (b) A histogram (heat map) $F_\nu^{\Delta,\xi}(z, \ell)$ of the fragment length $\ell$ of the ATAC-seq data at position $z$ from the TSSs as defined in (6). (left) CD4$^+$T, (center) ATL, and (right) HAM. The graphs are plotted for $z \in \{n\Delta \mid -37 \leq n \leq 37\}$ and $\ell \in \{n\xi \mid 0 \leq n \leq 49\}$, where $\Delta = 1000/75$ and $\xi = 10$.

These observations suggest that the statistics of open chromatin regions and nucleosome positioning around the TSSs in ATL cases is distinct from HAM cases and healthy CD4$^+$T cells, both of which showed more similar forms again.

## Giant sample-to-sample fluctuations of chromatin accessibility in ATL

We continue to elaborate on characteristic behaviors of ATL cells distinct from those of CD4$^+$T cells and HAM cells. To examine the distribution of open chromatin regions, we applied a systematic clustering algorithm that robustly detects open chromatin regions relevant to classifying the immunophenotype [15].

This algorithm characterizes open/closed chromatin regions in the following way. First, MACS2 uses ATAC-seq reads data for a given sample $s$ as the input and outputs a collection of peak data including the locations of peaks with their $p$-values. These peaks are considered as open regions of chromatin for sample $s$. The peaks are ordered in ascending $p$-values, and the first $M$ peaks are taken. The set of $M$ peaks are written as $\hat{g}_s^M = ((\gamma_k, \alpha_k, \beta_k), p_k)_{k=1}^M$, where the location of the $k$-th peak is the region $(\alpha_k, \beta_k)$ in the $\gamma_k$-th chromosome, and $p_k$ is the $p$-value of the $k$-th peak. Then, the set of the top $M = 64000$ peaks is determined as the optimal set of open chromatin regions, which effectively classifies the immunophenotypes of the samples [15]. In this procedure, peaks with high $p$-values (unreliable peaks) can be treated as noise and ignored for later analysis.

First, we tried to capture the genomic positions where chromatins tend to be open for at least one of the ATL, HAM, and healthy CD4$^+$T types. To quantify such chromatin regions, we constructed a new reference set $\hat{g}_0$ of peaks as follows: we concatenated all the reads from all the samples with the same cell type. Then, we used MACS2 to obtain a set of peaks for a cell type of the concatenated reads. Finally, we merged all the peaks obtained from the three cell types. For the explicit construction of the reference set $\hat{g}_0$, see Materials and methods. Next, we classified the reference set $\hat{g}_0$ into the set of all peaks overlapping gene-coding regions, which we denoted as $\tilde{g}_c$, and the set of all peaks overlapping with non-coding regions, which we denoted as $\tilde{g}_{nc}$.

To quantify open chromatin regions characterized by the peaks $\hat{g}_s^M$ of sample $s$ in each $\kappa$-th peak $(\alpha_\kappa[\tilde{g}], \beta_\kappa[\tilde{g}])$ from the reference set $\tilde{g} \in \{\tilde{g}_c, \tilde{g}_{nc}, \tilde{g}_c \cup \tilde{g}_{nc}\}$, we computed the width of overlapped peaks $O((\alpha_\kappa[\tilde{g}], \beta_\kappa[\tilde{g}]), \hat{g}_s^M)$ between $\tilde{g}$ and $\hat{g}_s^M$ in the peak location $(\alpha_\kappa[\tilde{g}], \beta_\kappa[\tilde{g}])$. We set $M = 64000$ as the provisionally optimal number for the immunophenotype classification [15]. Then, we focused on the average and variance of $O((\alpha_\kappa[\tilde{g}], \beta_\kappa[\tilde{g}]), \hat{g}_s^M)$. For details of the calculations, see Materials and methods.

As shown in Fig 4, healthy CD4$^+$T, HAM, and ATL cells showed similar behaviors at large average widths of the overlapped peaks. For small average widths, CD4$^+$T showed fewer peaks, compared with ATL cases. Additionally, as shown in Fig 5a, we found that healthy CD4$^+$T and HAM cases had similar sample-to-sample fluctuations, and ATL cases had a higher frequency at variances larger than $10^5$. As shown in Fig 5b and 5c, the larger sample-to-sample fluctuations in the ATL cases were found in both non-coding regions and coding regions. In contrast, Fig 5b shows apparent gaps between ATL cases and the other two cases at intermediate variances only in coding regions; ATL cases had a higher frequency at intermediate variances around $10^3$ compared with the other two cases.

The above analysis does not clearly distinguish healthy CD4$^+$T and HAM cells, in particular, with respect to the sample-to-sample fluctuations of the open chromatin regions. Therefore, we mixed the datasets of 5 cases of HAM cells and 5 cases of healthy CD4$^+$T cells to give set $\mathbb{S}$. As shown in Fig 5d, the variance in the mixed dataset was larger than of HAM cells or healthy CD4$^+$T cells alone. This finding indicates the distributions of open chromatin regions are different between healthy CD4$^+$T and HAM cells, although the two distributions showed similar sample-dependence. As a comparision, Fig 5a and 5d show that the sample-to-sample fluctuations of ATL cells are larger than the fluctuations of the mixed data except for the tail, where the samples are scarce.

Thus, ATL cases have a higher frequency at larger sample-to-sample fluctuations at the whole genome level and a higher frequency at intermediate sample-to-sample fluctuations only in coding regions. On the other hand, the chromatin structures in HAM cases show less sample-dependence, which is similar to CD4$^+$T cells, implying the existence of a certain trend common to all the HAM samples.

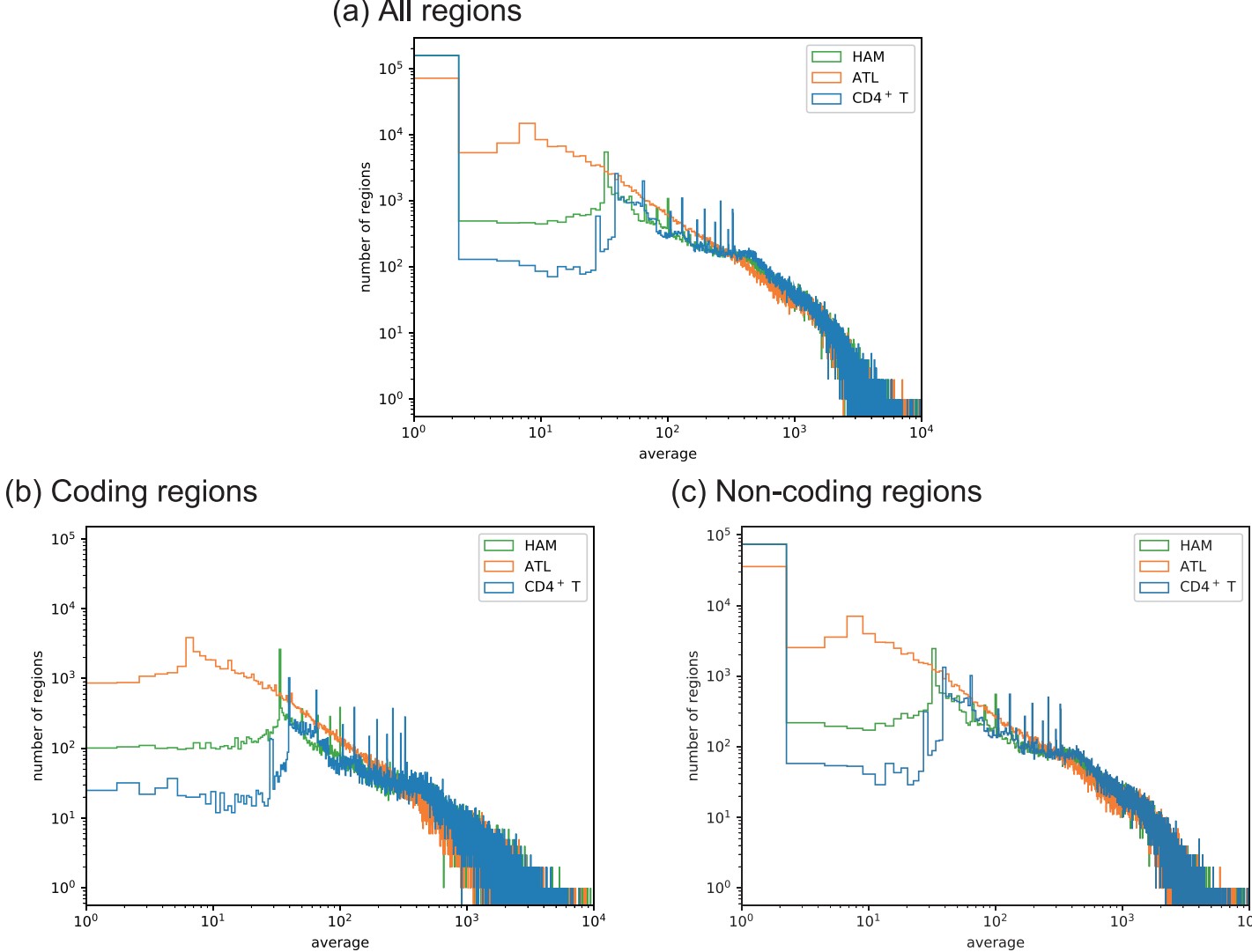

**Fig 4. Histograms $F_{\tilde{g},\mathbb{S}}^{(1)}(O;\Delta)$ of the averaged width $O$ of peaks overlapping with the reference open region as defined in (14) with bin width $\Delta$.** $\mathbb{S}$ consists of 6 HAM samples, 29 ATL samples, and 5 healthy CD4$^+$T samples. (a) All regions with $\tilde{g} = \tilde{g}_\text{c} \cup \tilde{g}_\text{nc}$. (b) Coding regions with $\tilde{g} = \tilde{g}_\text{c}$. (c) Non-coding regions with $\tilde{g} = \tilde{g}_\text{nc}$.

## mRNA: Distinctively expressed histone modifications in ATL

To analyze gene expressions, we examined the RNA-seq data of ATL cases.

We analyzed the gene expression pattern of HTLV-1-infected CD4$^+$ cells obtained from the peripheral blood of 4 ATL cases classified into acute subtype and of 4 healthy CD4$^+$T cells. The read counts of the RNA-seq data was normalized by TMM normalization [22] and used as the input data. For technical details, see section Materials and methods.

As a first step, we show a heatmap of gene expressions in Fig 6a, which implies the existence of certain difference between healthy CD4$^+$T cells and ATL cells. In order to elaborate on such elusive behaviors, as shown in Fig 6b, we performed principal component analysis (PCA) showing that the gene expression patterns differ significantly between ATL cells and healthy CD4$^+$ T cells. In ATL cells, there were 1289 genes up-regulated based on the condition of $\log_2 \text{FC}_i^\text{B} > 3$ and $p$-value $< 0.01$ and 944 genes down-regulated based on the condition of

## (a) All regions

## (b) Coding regions

## (c) Non-coding regions

## (d) All regions

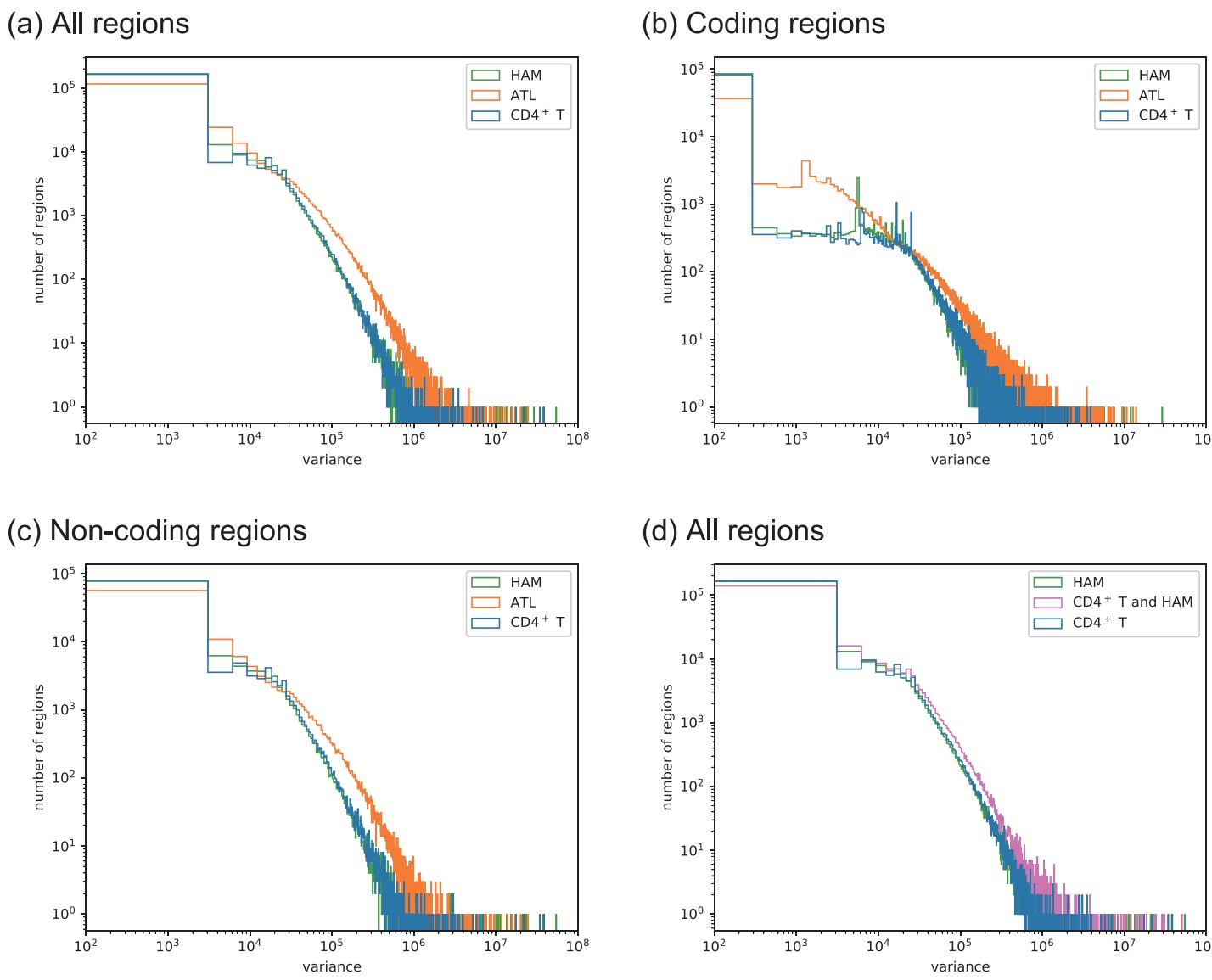

**Fig 5. Histograms $F_{\tilde{g}, \mathbb{S}}^{(2)}(O; \Delta)$ of the variance in width $O$ of peaks overlapping the reference open region as defined in (15) with bin width $\Delta$. $\mathbb{S}$ consists of 6 HAM** samples, 29 ATL samples, and 5 CD4$^+$T samples for (a), (b), and (c). (a) All regions with $\tilde{g} = \tilde{g}_c \cup \tilde{g}_{nc}$. (b) Coding regions with $\tilde{g} = \tilde{g}_c$. (c) Non-coding regions with $\tilde{g} = \tilde{g}_{nc}$. (d) All regions with $\tilde{g} = \tilde{g}_c \cup \tilde{g}_{nc}$ for CD4$^+$ T only, HAM only, and a mixed set of CD4$^+$ T and HAM with 5 samples for each cell type.

$\log_2 \text{FC}_i^B < -3$ and $p$-value $< 0.01$, where $\text{FC}_i^B$ is the fold change of the gene expression, for gene $i$, of ATL cells relative to healthy CD4$^+$T cells. For the details of log-fold change, see (2) and Materials and methods.

In addition, as shown in Fig 6d, Gene Ontology (GO) analysis using enrichR [23] revealed that in ATL cases, the up-regulated genes are enriched in histone modifications. Note that the combined scores of the genes with down-regulated expression are lower than those of up-regulated genes enriched for histone modification. Further, as shown in Fig 6c, many histone-related genes, such as *HIST1H2AH*, *HIST1H3C*, and *HIST1H4C*, are significantly up-regulated in ATL cases, where all genes beginning with HIST in the first 4 letters are regarded as histone-related genes.

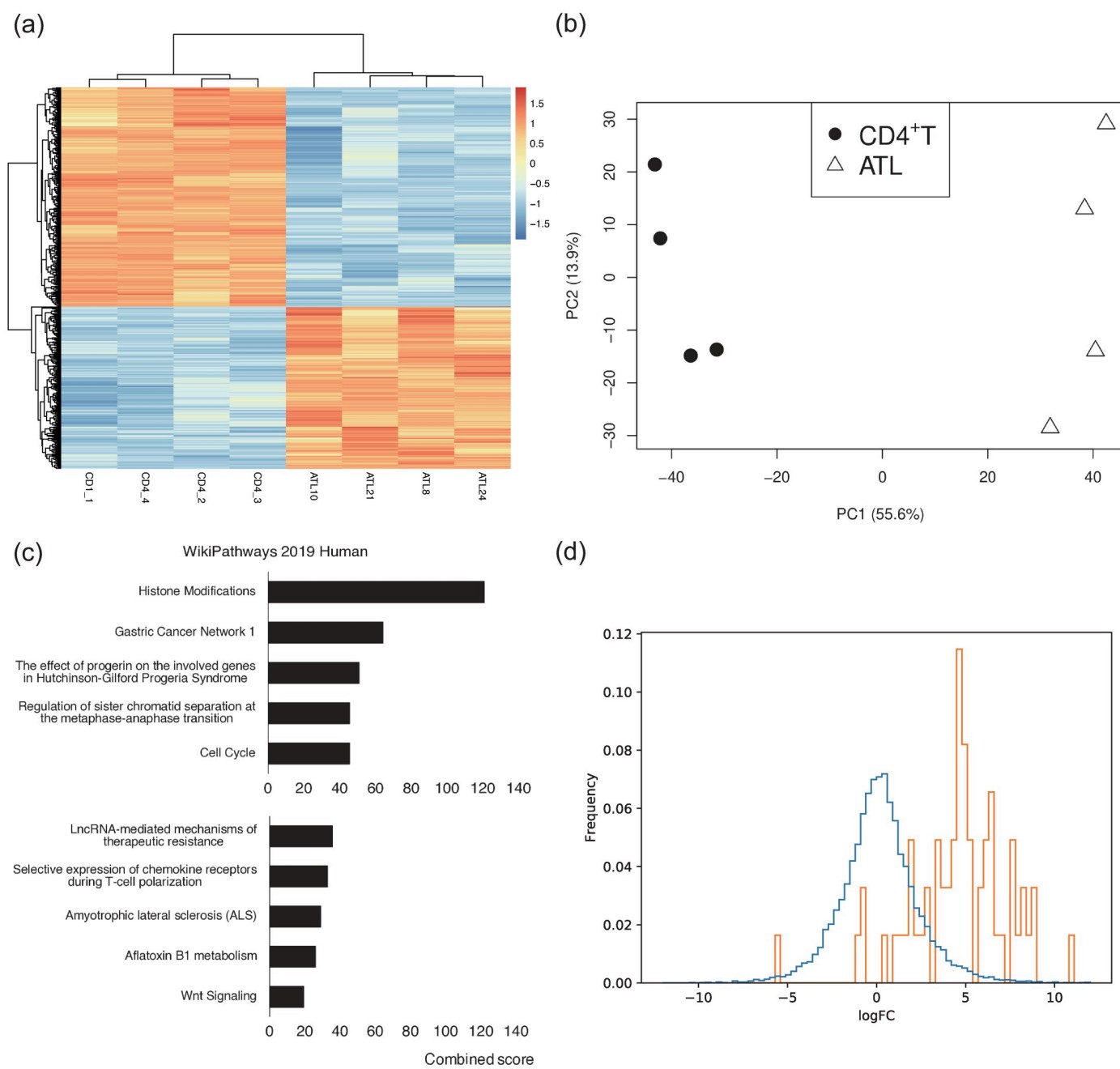

**Fig 6.** RNA-seq statistics: (a) A heatmap of gene expression. (b) PCA of RNA-seq for healthy CD4+T cells and ATL cells, where the percentages are the first and second contribution ratios. (c) GO analysis of RNA-seq for detecting the top 5 up-regulated gene expressions (top) and the top 5 down-regulated expressions (bottom) in terms of cell function for ATL. (d) A histogram of the log-fold change calculated in (18) and (19) between the gene expressions (RNA-seq) of ATL cases and of healthy CD4+T cells for all genes (blue line) vs. 61 histone-related genes (orange line). To select relevant genes, first we chose 69 genes whose names start with HIST. Then, we removed 8 genes, including 7 genes for which healthy CD4+T cells had no peak and 1 gene for which ATL cells had no peak.

## Correlation between chromatin accessibility and mRNA in ATL: Large change in mRNA expressions of the genes with nearly unchanged chromatin

The observations in previous sections led us to consider a correlation between the anomalous chromatin properties and the anomolous gene expression levels in ATL cells. To gain a direct

quantification of how chromatin structures are correlated with gene expressions in ATL cells, we performed an integrated analysis of ATAC-seq and RNA-seq data for ATL cells mainly in comparison with CD4$^+$T.

First, we quantify the difference between a set of peaks obtained from ATAC-seq for cell type $t_1$ and that for cell type $t_2$. In order to realize it, as in Figs 4 and 5, we used our algorithm to classify the top $M$ peaks as open chromatin regions [15]. Then, for a given gene $i$, we computed $\overline{A}_i(t_1)$ defined by the average width of peaks overlapping with gene $i$ among samples with type $t_1$. Then, we calculated the fold change of the quantity computed above for samples with type $t_1$ compared to samples with type $t_2$, which is explicitly written as

$$\mathrm{FC}_i^{\mathrm{A}}(t_1, t_2) := \frac{\overline{A}_i(t_1)}{\overline{A}_i(t_2)}. \tag{1}$$

The logarithm of the fold change, which is $\log_2 \mathrm{FC}_i^{\mathrm{A}}(t_1, t_2)$, corresponds to the horizontal coordinate $x = x_i$ of the point for each gene $i$ in Fig 7a and 7c.

Second, we used the RNA-seq data to quantify the expression of each gene $i$ for given samples. We computed $\overline{B}_i(t)$ defined by the average of the normalized read counts of the RNA-seq data, which is realized by edgeR [24, 25], in gene $i$ over all samples with type $t$. We calculated the fold change of the expression of gene $i$ for type $t_1$ compared to that for type $t_2$ as

$$\mathrm{FC}_i^{\mathrm{B}}(t_1, t_2) := \frac{\overline{B}_i(t_1)}{\overline{B}_i(t_2)}. \tag{2}$$

The logarithm of the fold change of the gene expression, which is $\log_2 \mathrm{FC}_i^{\mathrm{B}}(t_1, t_2)$, corresponds to the vertical coordinate $y = y_i$ of the point for gene $i$ in Fig 7a and 7c.

The heatmap in Fig 7b and 7d corresponds to the two-dimensional histogram in terms of polar coordinate system $(R, \theta)$ of the scatter plots of the points $(x, y)$; namely, $x = R \cos \theta$ and $y = R \sin \theta$. Recall that $M$ is originally determined as $M = 64000$ such that the clustering of the ATAC-seq samples is closest to the appropriate immunophenotype [15]; we will basically set $M = 64000$. Note that the points which have zero as a value of $\overline{A}_i$ or $\overline{B}_i$ are excluded in the scatter plots and the histograms (see also Table 2). For details of the calculations, see section Materials and methods. It should be noted that if the distribution function in $x - y$ coordinate system is isotropic, the distribution function in the polar coordinate system is uniform toward $\theta$-direction; in otherwords, the color scale is flat toward $\theta$-direction.

As shown in Fig 7a for $t_1 = $ ATL and $t_2 = $ CD4$^+$T, it might be not quite obvious to judge how the scatter plot is anisotropic. However, as shown in Fig 7b, the two-dimensional histogram in the polar coordinate system help us to capture more clear anisotropic structure; in particular, there seems to be two peaks around $\theta$ equal to $\pm90$. Indeed, $\theta = \pm90$ corresponds to the case that chromatin structure is completely unchanged.

Note that as shown in Fig 7c and 7d, the case with $M = \infty$, where all peaks from ATAC-seq are included in the analysis, shows less anisotropic in the window between $-90 < \theta < 90$ compared to that of the case with $M = 64000$; Thus, two-peaks-like structure found for $M = 64000$ is rather masked for $M = \infty$. It implies that the chromatin structures detected by the top $M = 64000$ peaks effectively capture the elusive differences between CD4$^+$T and ATL cells from the viewpoint of the correlation between ATAC-seq and RNA-seq.

Next, in order to elaborate on the difference between ATL cells and CD4$^+$T in terms of the correlation between ATAC-seq and RNA-seq, we used the following set of cell types $\mathbb{T}$ as control types:

$$\mathbb{T} = \{\mathrm{HSC}, \mathrm{CD8}^+\mathrm{T}, \mathrm{Mono}\},$$

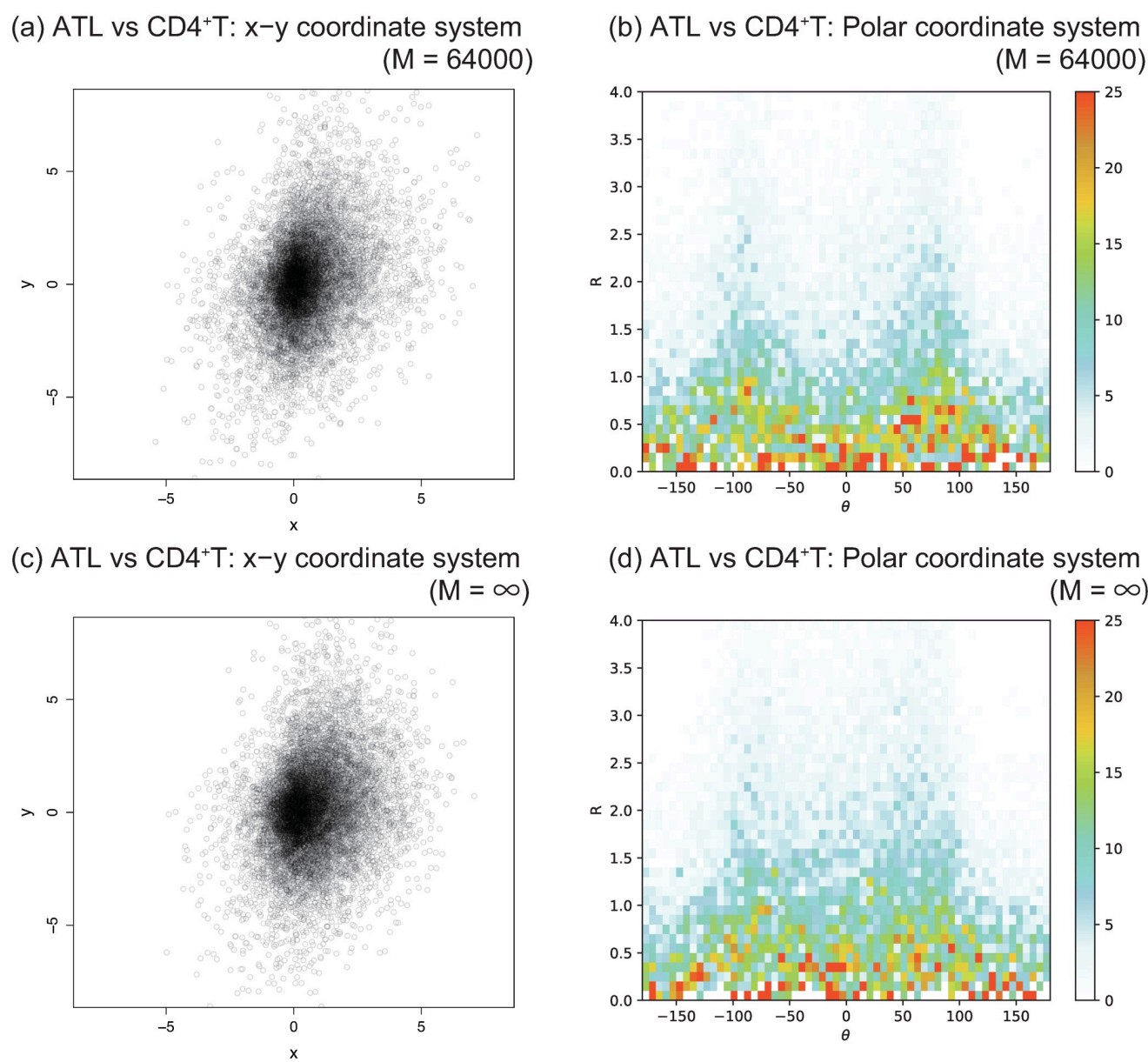

**Fig 7.** (a)(c) A scatter plot $\rho_{t_1,t_2}^M$ of log-fold changes comparing ATL samples versus CD4$^+$T data where $t_1 = $ ATL and $t_2 = $ CD4$^+$T. The horizontal coordinate $x$ is $\log_2$ FC$_i^A$, where FC$_i^A$ is defined in (1). The vertical coordinate $y$ is $\log_2$ FC$_i^B$ where FC$_i^B$ is defined in (2). (b)(d) The corresponding two-dimensional histogram of modified numbers of points by polar coordinates, for which distribution function is $\eta_{t_1,t_2}^M$. The coordinates $R$ and $\theta$ are a polar coordinate system of $x-y$ coordinate system such that $x = R\cos\theta$, $y = R\sin\theta$. The color scale is the number of points $(x, y)$ in the polar coordinate system within each bin divided by $R$. Bin width is 0.1 for $R$ and 6 for $\theta$. $M = 64000$ for (a) and (b). $M = \infty$ for (c) and (d).

where HSC and Mono are hematopoietic stem cell and monocyte, respectively. Concretely, we performed the same analysis taken above where $t_1$ is either ATL or CD4$^+$T and $t_2 \in \mathbb{T}$ with $M = 64000$. Note that the points which have zero as a value of $\overline{A}_i$ or $\overline{B}_i$ are excluded in the histograms (see also Table 2).

Fig 8 show that ATL cells have exceptionally distinct structures in terms of the correlation between chromatin accessibility and mRNA compared with CD4$^+$T cells. In particular, as

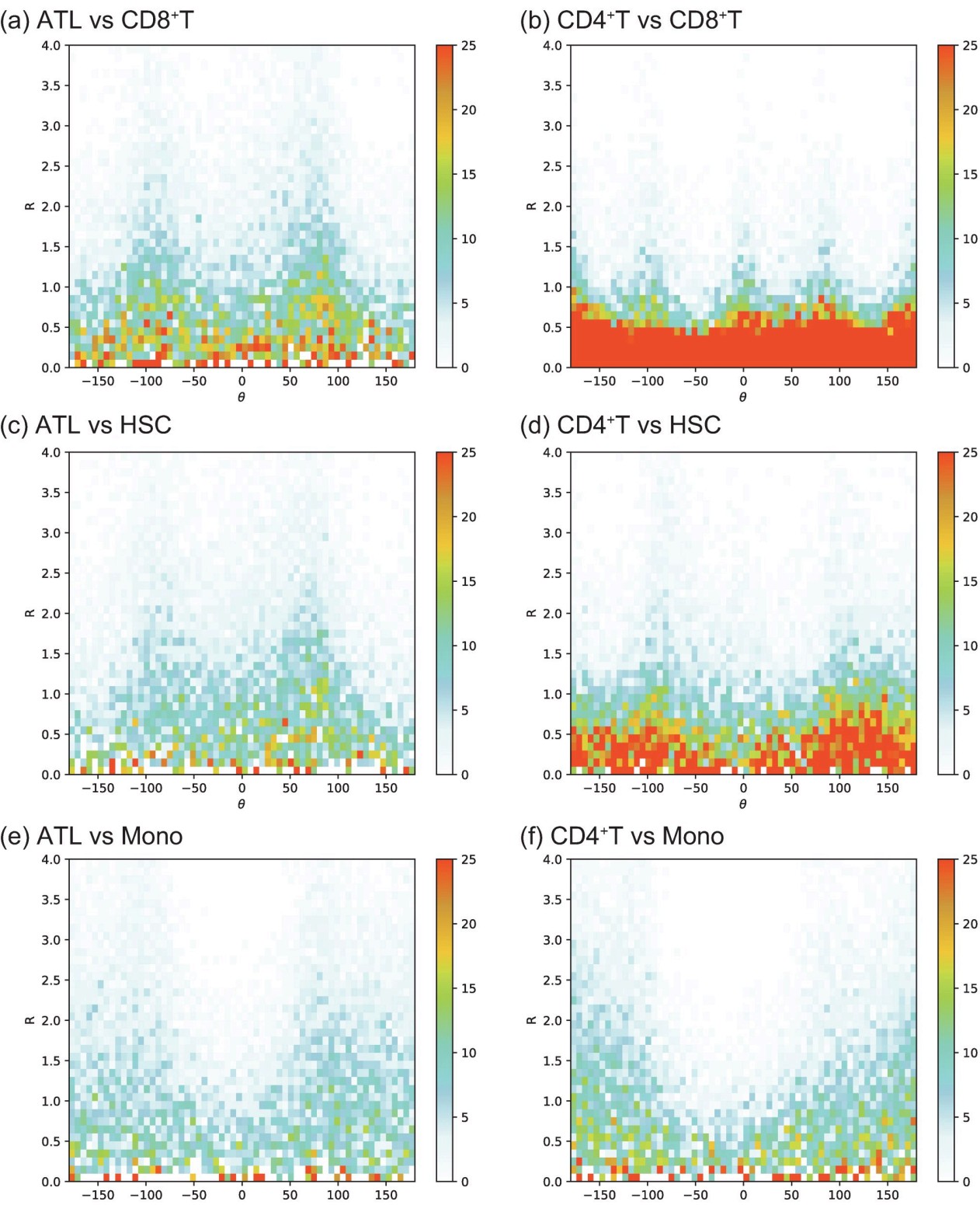

**Fig 8. The histogram of distribution function $\eta_{t_1,t_2}^M$ with bin width of $R$ and $\theta$ equal to 0.1 and 6, respectively. $M$ = 64000.** (a) $(t_1, t_2)$ = (ATL, CD8+T). (b) $(t_1, t_2)$ = (CD4+T, CD8+T). (c) $(t_1, t_2)$ = (ATL, HSC). (d) $(t_1, t_2)$ = (CD4+T, HSC). (e) $(t_1, t_2)$ = (ATL, Mono). (f) $(t_1, t_2)$ = (CD4+T, Mono).

**Table 2. Number of classified genes by ATAC-seq and RNA-seq with $M$ = 64000.** $N_{00}$ corresponds to the number of the genes, for which gene expression is zero and the peak width inside is zero. $N_{01}$ corresponds to the number of the genes, for which gene expression is zero and the peak width inside is nonzero. $N_{10}$ corresponds to the number of the genes, for which gene expression is nonzero and the peak width inside is zero. $N_{11}$ corresponds to the number of the genes, for which gene expression is nonzero and the peak width inside is nonzero. Only the data of genes counted in $N_{11}$ for both of two cell types $(t_1, t_2)$ are plotted in Figs 7 and 8. Note that zero or nonzero mean zero or nonzero on the average of all the analyzed samples with a same cell type.

| Type | $N_{00}$ | $N_{01}$ | $N_{10}$ | $N_{11}$ |
|------|------|------|------|------|
| ATL | 6409 | 1023 | 7762 | 13316 |
| CD4$^+$T | 7731 | 1724 | 7220 | 11835 |
| CD8$^+$T | 8101 | 1879 | 6761 | 11769 |
| HSC | 7610 | 1554 | 7391 | 11955 |
| Mono | 7895 | 2253 | 6138 | 12224 |

shown in Fig 8a and 8c, in a similar manner to that found in the comparison between ATL and CD4$^+$ T, we found again two-peaks-like structure around $\theta$ equal to ±90. Indeed, as shown in Fig 8b, 8d and 8f, the three cases with $t_1 = $ CD4$^+$T show various structures without such two-peaks-like structure, which depends on control types $t_2$. It seems that Fig 8e does not have clear two-peaks structure; however, one should see that compared to Fig 8f, the frequency with $\theta = $ ±180 is certainly decreased, which seem to make the form of the distribution function closer to two-peaks-like structure.

Summarizing the above finding, compared to CD4$^+$T, ATL cases robustly show a tendency that the genes, of which chromatin part is nearly unchanged, in the analyzed class of genes is largely affected in terms of gene expression.

## Classification of ATL and HAM samples based on chromatin structures

We next examined ATL cases by inferring the past cell status before infection with HTLV-1 and the current cell status in terms of immunophenotypes compared with normal hematopoietic cells.

It remains unclear why most ATL cells have an immunophenotype similar to CD4$^+$ memory T cells even though HTLV-1 infects multiple hematopoietic cells [26]. Indeed, it was reported that ATL samples showed CD4 and CD45RO positive, which implies that the onset of ATL is strongly related to this subclass of T cells [19]. In order to look at this phenomenon from the viewpoint of chromatin structures, we ask whether chromatin structures of T-cell Receptor Excision Circles (TREC) region, which is removed during T-cell receptor gene rearrangement, are open. If T-cell receptor gene rearrangement occurs, we expect to observe no reliable peak in TREC region for ATL samples except for certain peaks due to noise effects induced by uncontrollable experimental procedures. Those reliable peaks are estimated as the top 64000 peaks.

Specifically, using ATAC-seq data from ATL cells and 13 human primary blood cell types from healthy donors, we computed the total width of peaks in the TREC region. As shown in Fig 9, the total width of peaks in the TREC region in ATL/HAM cells is less than that of CD8$^+$T cells on the average, whereas the average total width of each of other hematopoietic cell types is larger than that of CD8$^+$T cells. Further, we computed the ratio of the total width of all the peaks in TREC region to that of *TRA* gene coding region. Note that *TRA* gene coding region includes TREC region. Then, we observed that the ratio for CD4$^+$T, CD8$^+$T, HAM, and ATL on the sample average is very close to 0 whereas that for all the other cell types is larger

## (a) TREC region

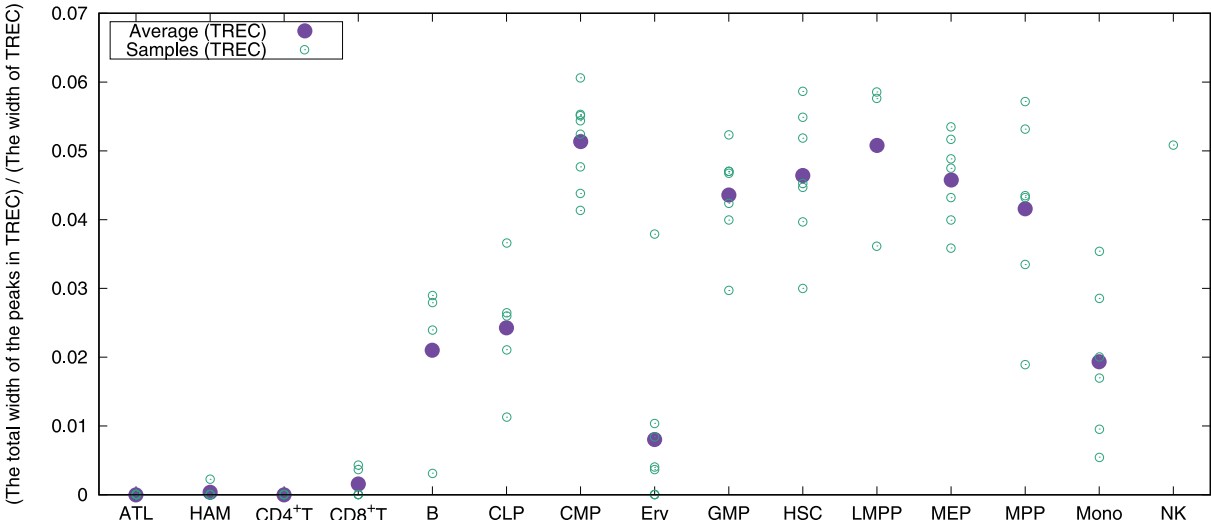

## (b) TREC region vs. TRA gene region

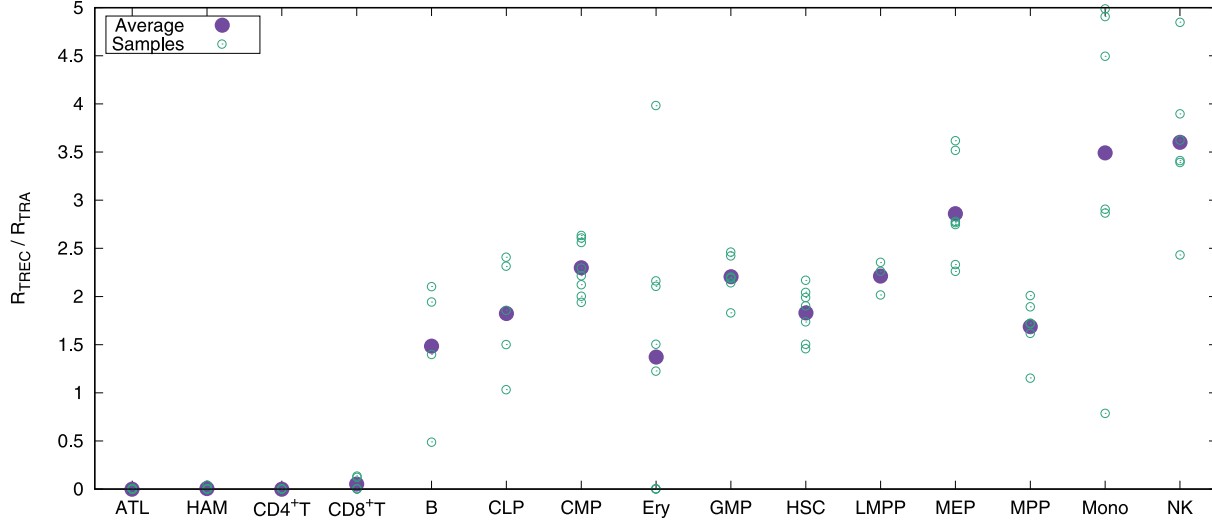

**Fig 9.** (a) The ratio of the total width of all the peaks regions overlapping with TREC region to the width of TREC region. It is explicitly defined as $R_{\text{TREC}}(s) := O((\alpha_{\text{TREC}}, \beta_{\text{TREC}}), \hat{g}_s^M)/W_{\text{TREC}}$ for a sample $s$, where $(\alpha_{\text{TREC}}, \beta_{\text{TREC}})$ is the location of TREC region, $W_{\text{TREC}} = \beta_{\text{TREC}} - \alpha_{\text{TREC}}$, and $M$ is 64000. (See also Materials and methods.) The ratio takes values from 0 to 1. (b) The ratio $R_{\text{TREC}}(s)/R_{\text{TRA}}(s)$, where $R_{\text{TRA}}(s)$ is the ratio defined for TRA region in the same way as that for TREC region. The widths of regions are $W_{\text{TRA}} = 931018$ and $W_{\text{TREC}} = 89008$ for TRA gene coding region and TREC region, respectively. Note that for all the samples of ATL and CD4$^+$T, there are no peaks in TREC region. Additionally, 1 sample of HAM, 2 samples of CD8$^+$T, and 3 samples of Ery have no peaks in the TREC region. Other samples have at least one peak in the TREC region, and all the samples have at least one peak in the TRA region. The average of the ratio in TREC region for NK is 0.118···, which is outside of the range shown in the figure.

than 1. Thus, this is supporting evidence, in terms of chromatin levels, that the infected cells of the ATL/HAM patients used in the analysis were differentiated into T cells.

To investigate the effect of large-scale variations of chromatin structures on the classification, we used our algorithm to quantitatively evaluate differences between ATL cells and hematopoietic cells from healthy donors [15]. Using this algorithm, we calculated the

**Table 3. Clustering results of ATAC-seq from ATL cases in terms of immunophenotypes using the method in [15].** The data corresponding to accession numbers can be downloaded from DNA Data Bank of Japan (DDBJ).

| Sample labels | Accession number | first | second | third | Clinical subtype |
|---|---|---|---|---|---|
| ATL1 | DRR574504 | Ery | Mono | CLP | Acute |
| ATL2 | DRR574505 | Mono | CD4$^+$T | B | Chronic |
| ATL3 | DRR574506 | CD4$^+$T | CD8$^+$T | NK | Chronic |
| ATL4 | DRR574507 | CD4$^+$T | CD8$^+$T | B | Acute |
| ATL5 | DRR574508 | Mono | Ery | CD4$^+$T | Chronic |
| ATL6 | DRR574509 | Ery | Mono | B | Chronic |
| ATL7 | DRR574510 | CD4$^+$T | CD8$^+$T | B | Acute |
| ATL8 | DRR574511 | CD4$^+$T | CD8$^+$T | NK | Acute |
| ATL9 | DRR574512 | CD4$^+$T | CD8$^+$T | B | Acute |
| ATL10 | DRR574513 | CD4$^+$T | CD8$^+$T | B | Acute |
| ATL11 | DRR574514 | CD4$^+$T | CD8$^+$T | B | Acute |
| ATL12 | DRR574515 | CD4$^+$T | CD8$^+$T | B | Acute |
| ATL13 | DRR574516 | CD4$^+$T | CD8$^+$T | B | Acute |
| ATL14 | DRR574517 | CD4$^+$T | CD8$^+$T | B | Acute |
| ATL15 | DRR574518 | CD4$^+$T | B | CD8$^+$T | Acute |
| ATL16 | DRR574519 | CD4$^+$T | CD8$^+$T | NK | Acute |
| ATL17 | DRR574520 | Ery | Mono | B | Chronic |
| ATL18 | DRR574521 | CD4$^+$T | CD8$^+$ T | B | Acute |
| ATL19 | DRR574522 | CD4$^+$T | B | CD8$^+$T | Acute |
| ATL20 | DRR574523 | CD4$^+$T | CD8$^+$T | B | Chronic |
| ATL21 | DRR574524 | CD4$^+$T | B | CD8$^+$T | Acute |
| ATL22 | DRR574525 | CD4$^+$T | B | CD8$^+$T | Acute |
| ATL23 | DRR574526 | Ery | CD4$^+$T | Mono | Acute |
| ATL24 | DRR574527 | CD4$^+$T | CD8$^+$T | B | Acute |
| ATL25 | DRR574528 | CD4$^+$T | CD8$^+$T | B | Acute |
| ATL26 | DRR574529 | CD4$^+$T | CD8$^+$T | B | Acute |
| ATL27 | DRR574530 | CD4$^+$T | CD8$^+$T | B | Chronic |
| ATL28 | DRR574531 | CD4$^+$T | CD8$^+$T | B | Acute |
| ATL29 | DRR574532 | CD4$^+$T | CD8$^+$T | B | Acute |

Hamming distances between the peak-based binarized genome of ATL cells and of hematopoietic cells from healthy donors. For this purpose, we used ATL samples and 77 ATAC-seq datasets from 13 human primary blood cell types. As summarized in Table 3, the majority of ATL samples are close to CD4$^+$T cells, as expected by the above analysis about the past cell status. We also found that the ATAC-seq patterns of some ATL cases are close to myeloid cells such as erythroid cells and monocytes. Note that by using the same method as that for ATL samples, ATAC-seq data of all the HAM samples are classified into CD4$^+$T as the first, CD8$^+$T as the second, NK as the third. Thus, the classification of the HAM samples by chromatin structures does not depend on the samples.

Looking at the ATAC-seq data further, a footprint analysis for the identification of differential motifs revealed that ETS1, IRF2, and RUNX2 had deeper footprints and higher DNA accessibility at the flanking locations of their motifs in healthy CD4$^+$T cells, while NRF1, KLF4, and KLF9 had deeper footprints in ATL cells (Fig 10) [27]. These observations suggest that transcription factors such as NRF1, KLF4, and KLF9 play an important role in ATL.

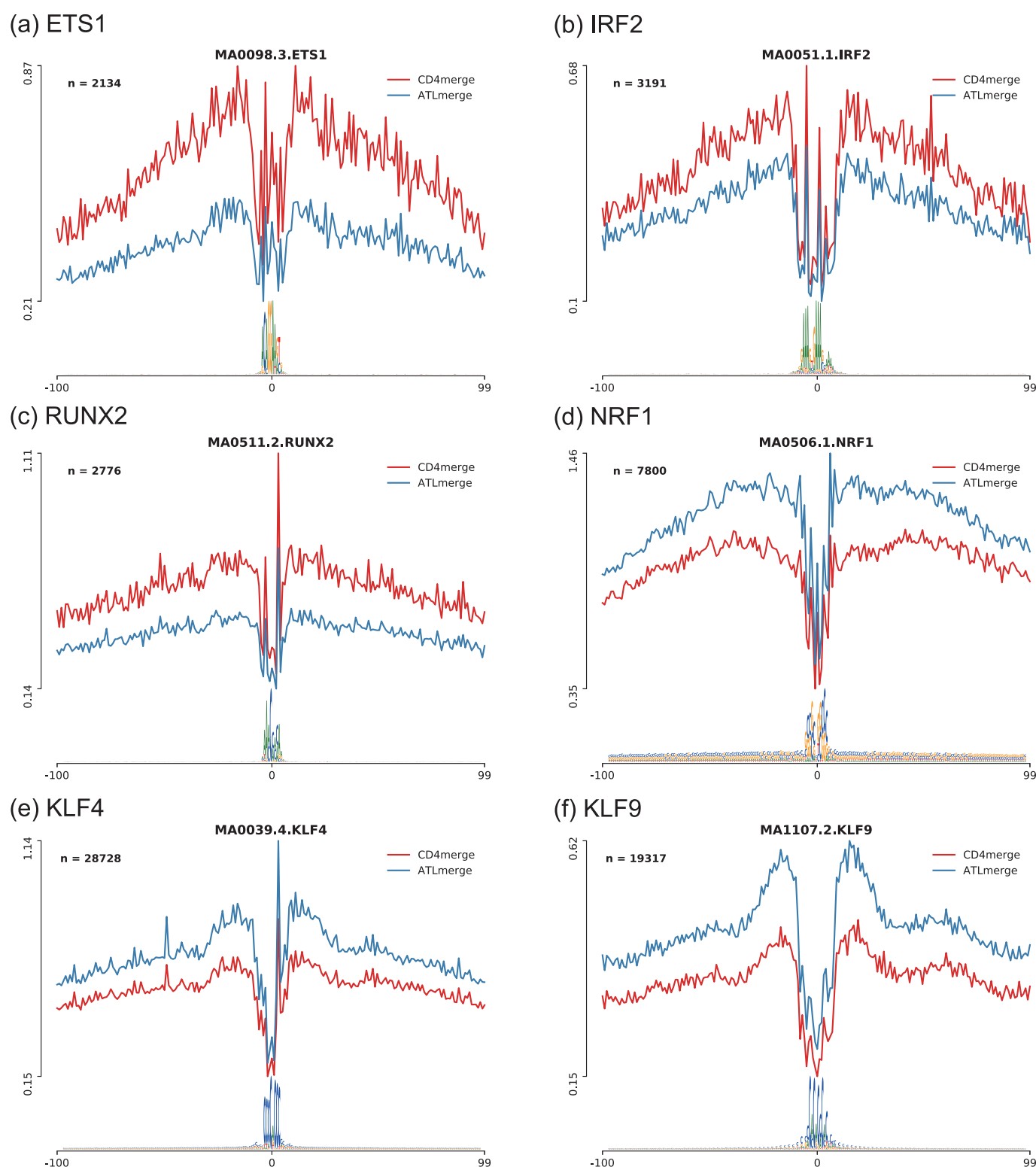

**Fig 10. A footprint analysis of transcription factors.** (a)-(f) Distances from motifs (horizontal axis) vs. the averaged number of reads over all parts of a given motif (vertical axis) outputted from HINT-ATAC [27].

**Table 4. TPM means the values of the transcripts per kilobase million (TPM) output from RNA-seq through RSEM algorithm.** The data corresponding to accession number can be downloaded from DBBJ.

| Sample labels | Accession number | TPM of isoform 1 | TPM of isoform 2 |
|---|---|---|---|
| ATL8 | DRR573818 | 0.19 | 0 |
| ATL10 | DRR573819 | 11.43 | 1.31 |
| ATL21 | DRR573820 | 1.94 | 0 |
| ATL24 | DRR573821 | 0.65 | 0.06 |
| The average | | 3.55 | 0.34 |

To ascertain whether the mRNA expression in the ATL cells reflects the characteristics of myeloid cells, we analyzed the RNA-seq data from healthy CD4$^+$ T cells and HTLV-1-infected CD4$^+$ cells from 4 ATL cases; The samples are ATL8, ATL10, ATL21, and ATL24 listed in Tables 3 and 4. We used the condition $\log_2$ FC > 1 and $p$-value <0.01 to identify up-regulated genes and found two candidates: CD71 (TFRC), which is ubiquitously expressed by erythroid precursors [28, 29], and KLF4, which is highly expressed in myeloid cells and essential for monocyte differentiation [30].

## Chromatin-based systematic selection of key genes in ATL

To identify genes that are specially expressed in ATL cells, we investigated whether there are characteristic open chromatin regions in gene coding regions that are common to all the ATL cells but not to hematopoietic cells derived from healthy individuals. Although this class of genes is not taken into account in the analysis with Figs 7 and 8, this class of genes potentially show unique properties of ATL cells.

We therefore compared the chromatin accessibility between 29 ATL samples and 77 ATAC-seq datasets from 13 human primary blood cell types [31]. We applied our algorithm to detect such key genes, using $M$ as a parameter for clustering the ATAC-seq data corresponding to immunophenotypes [15]. First, we defined a subset of regions of top $M$ peaks, where all ATL samples had peaks but no 13 human primary blood cell types from healthy donors had. We call this subset ATL-specific open regions. Second, we detected the genes in which at least one of ATL-specific open region was located for a given $M$.

As shown in Fig 11, the number of key genes was 0 at $M = 2000$, 1 at $M = 4000$, 3 to 4 between $M = 8000$ and 56000, and then dropped back to 1 for values larger than $M = 64000$. Among these genes, we picked those that stayed in intervals of $M \geq 8000$. Concretely, *TLL1* (Tolloid-like-1) gene appeared from $M = 8000$-16000, *EVC* (Ellis-van Creveld) gene and *CRMP1* (Collapsin response mediator protein 1) gene appeared from $M = 16000$-48000, *TNRC6A* (Trinucleotide Repeat Containing Adaptor 6A) gene appeared from $M = 32000$-48000, and *UST* (Uronyl-2-sulfotransferase) gene appeared from $M = 64000$. As a reference in Fig 12a, we show the the ATAC-seq peak distributions around *TLL1* gene. Note that the detected genomic region for *EVC* and *CRMP1* is the same because the two genes overlap. Thus, by our selection, the candidates determined as key genes are few.

Consistently, *EVC* has been reported to be overexpressed in ATL and plays an important role in cellular Hedgehog activation [32]. *UST* was also highly expressed in ATL cases, though the relationship between the function of *UST* and the symptoms of ATL has not been explicitly clarified [33]. On the other hand, to the best of our knowledge, *TLL1* and *TNRC6A* have not been studied in this context. While TLL1 is known to be necessary for normal septation and positioning of the heart [34], a more recent report found that it is associated with the development of hepatocellular carcinoma after the eradication of HCV [35]. Further, TLL1 is a

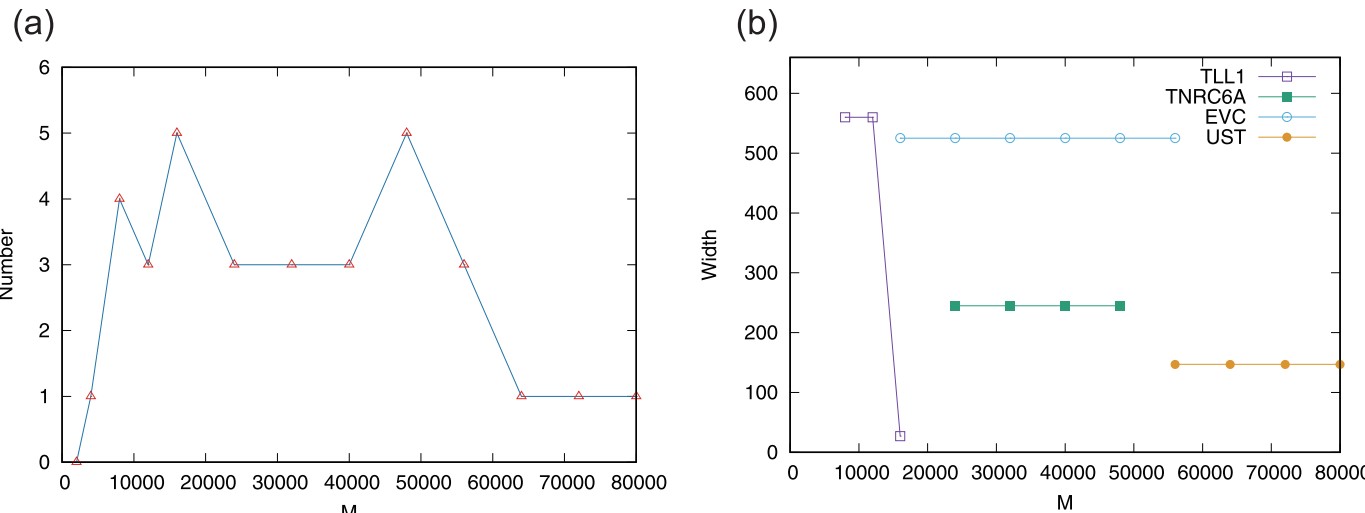

**Fig 11.** (a) The number of the genes which satisfies the two conditions: (i) the gene overlaps with, at least, a part of the top $M$ peaks obtained from ATAC-seq data of ATL samples, but (ii) does not overlap with all the top $M$ peaks of all the healthy hematopoietic cell samples. The data are taken from $M = 8000$ to $M = 80000$ by 8000 increments for the value of $M$ and additionally from $M = 4000, 12000$. (b) The width of all the peaks overlapping with the gene which stay selected after more than 8000 increments for the value of $M$.

member of the BMP1/TLD (bone morphogenetic protein1/tolloid)-like proteinase family; BMP1 controls latent TGF-$\beta$ activation via the cleavage of LTBP1 (latent TGF-$\beta$ binding protein-1) [36], and TGF-$\beta$ plays important roles in cancer progression [37]. Thus, we picked up *TLL1* as a promising candidate among genes expressing ATL-specific functions.

## TLL1 can strongly regulate TGF-$\beta$

We next considered the gene expression of *TLL1* in ATL cells and TLL1's effect on the maturation process of TGF-$\beta$ (See Materials and methods for the detail of the whole experimental set-ups). First, as shown in Fig 12c, we performed real-time PCR that showed TLL1 is expressed in ATL cases but not in the peripheral blood mononuclear cells of healthy donors. This result is consistent with the Human Protein Atlas, which reported that TLL1 mRNA is not detected in most adult tissues including immune cells or any hematopoietic lineage. By analyzing the RNA-seq data, we also confirmed that TLL1 mRNA is not expressed in normal hematopoietic cells, but it was expressed in all examined ATL cases. The same was not true for *TNRC6A* gene, which, according to RNA-seq data, did not show any systematic expression change between ATL cases and normal hematopoietic cells. Indeed, it turns out that *TLL1* corresponds to the gene, which is counted in $N_{11}$ for ATL and in $N_{01}$ for 13 human primary blood cell types with $M = 64000$ (See also Table 2 and Fig 11b.)

Next, we considered the relationship between TLL1 and the maturation process of TGF-$\beta$. TLL1 has two mRNA isoforms: TLL1 isoform 2 (NM_001204760) lacks many exons from the 3' end of TLL1 isoform 1 (NM_012464). Thus, we asked if both isoforms regulate TGF-$\beta$ in a manner similar to BMP1.

To investigate this possibility, as shown in Fig 13a, we performed a luciferase assay using the pre-mature form of TGF-$\beta$ co-expressed with either TLL1 isoform 1 or 2 in a HepG2 cell line, which is a TGF-$\beta$ responsive cell line [38]. As shown in Fig 13b, we found that compared to the sample without TLL1 (Case 2), TLL1 isoform 1 (Case 3) activates the pre-mature form of TGF-$\beta$ for maturation, but TLL1 isoform 2 (Case 4) represses the maturation. It should be

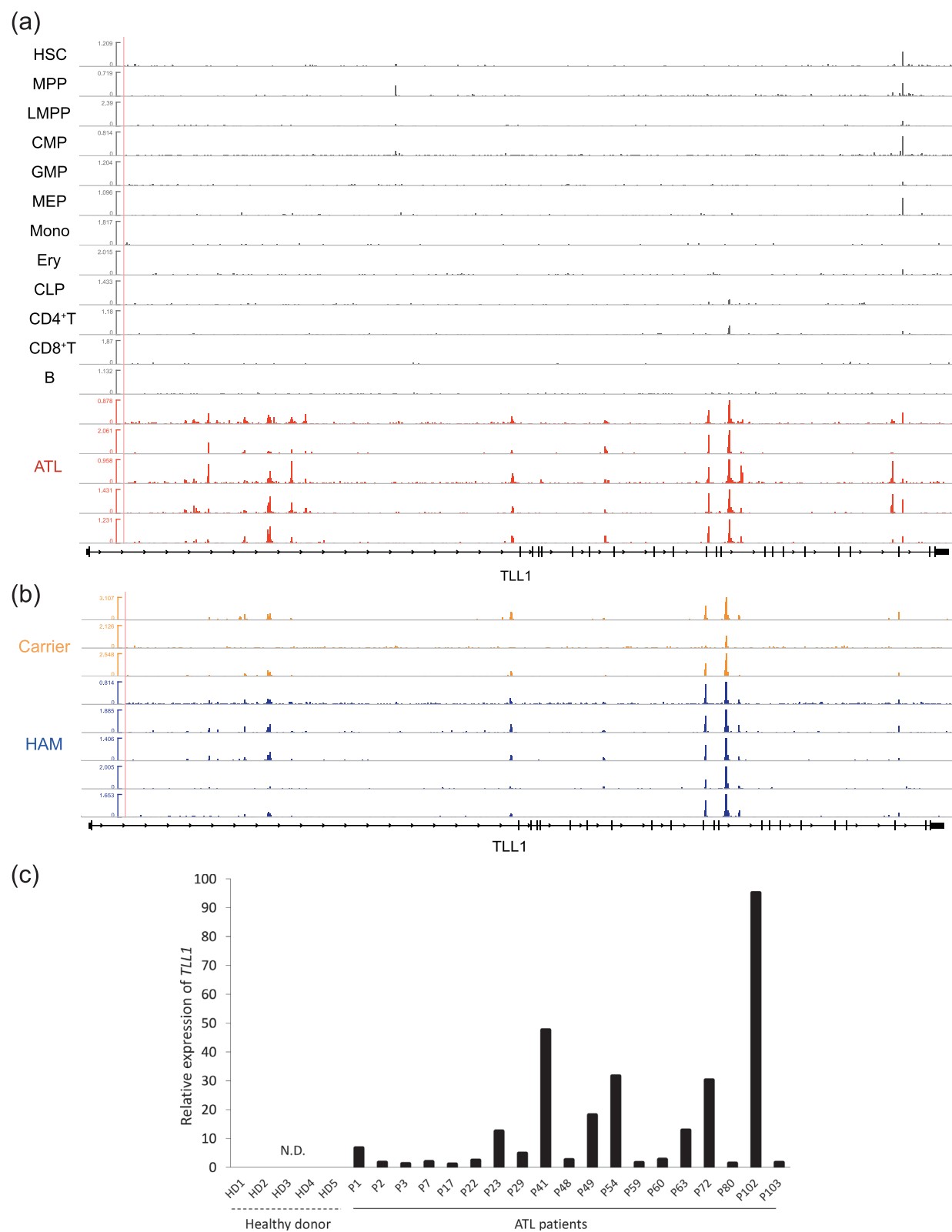

**Fig 12.** (a,b) Chromatin accessibility around the TLL1 region visualized by Integrative Genomics Viewer. (c) Expression of TLL1 relative to GAPDH (Glyceraldehyde-3-phosphate dehydrogenase) for ATL sample $s$. The relative expression of sample $s$ is defined as $2^{\Delta C_s^0}$, where $\Delta C_s^0$ is computed using the $\Delta\Delta$ Ct method; for details, see (21). N.D. stands for no detection of TLL1.

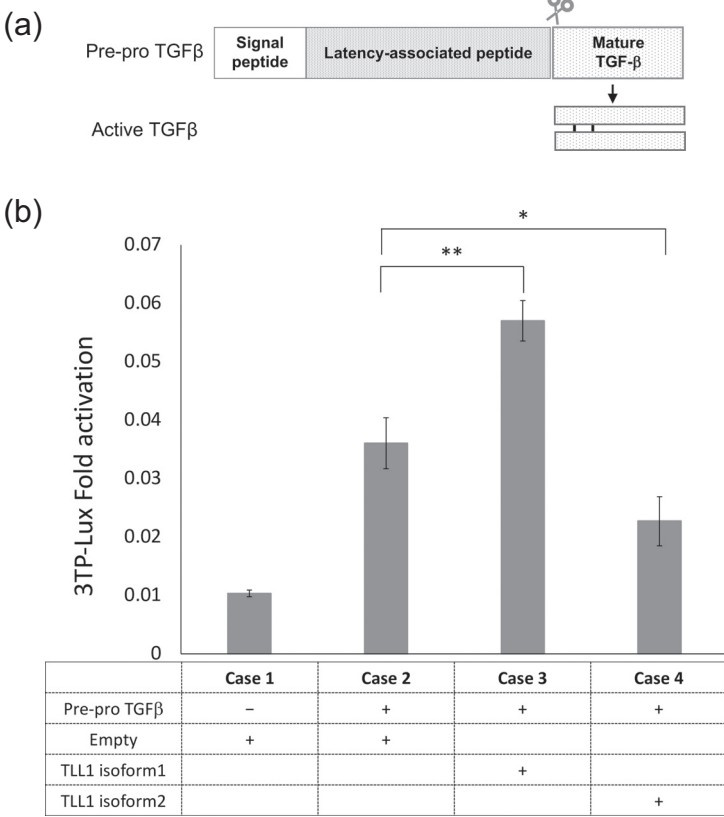

**Fig 13.** (a) Schematic picture of pre-pro TGF-$\beta$ and mature TGF-$\beta$. (b) TGF-$\beta$ activation was measured by 3TP-Lux protein activation, which depended on the TLL1 isoform type. The $p$-value is 0.0028 (**) for Case 2 vs. Case 3 and 0.019 (*) for Case 2 vs. Case 4 (t-test).

noted that the difference of luciferase activity between isoform 1 (Case 3) and isoform 2 (Case 4) with pre-mature TGF-$\beta$ approximated the difference between the TLL1-less condition with pre-mature TGF-$\beta$ (Case 2) and without pre-mature TGF-$\beta$ (Case 1). Thus, the results suggest that TLL1 is able to strongly regulate TGF-$\beta$ depending on the isoform expression ratio.

Taking into account the role of TGF-$\beta$ in cancer progression, the above observations suggest that TLL1 isoform 1 is more expressed in ATL samples, compared to TLL1 isoform 2. Indeed, as shown in Table 4, we analyzed again the RNA-seq of ATL samples and confirmed that the expression of TLL1 isoform 1 is about 10.3 times larger than that of TLL1 isoform 2 on the average over 4 samples (see Materials and methods for the detail). Note that TLL1 isoform 2 is not detected in 2 samples out of 4 samples.

Addtionally, to understand TLL1's function, we conducted experimental studies on MT-2, an ATL cell line that does not express TLL1. We prepared three samples of MT-2: one transduced with an empty vector, another with TLL1 isoform 1, and the third with TLL1 isoform 2. We analyzed the RNA-seq data from the three samples by computing the gene expression ratio between them.

Among the 22963 genes, we picked up the genes which have nonzero read counts in MT-2 transduced with an empty vector. A part of such genes showed significantly altered expressions depending on the type of isoform. As shown in Table 5, four genes were significantly up-regulated when TLL1-isoform 1 was transduced: CCR6 is related to the regulation of Treg

**Table 5. Relative expression (RE) of genes in TLL1-transduced MT-2 cells over MT-2 cells transduced with an empty vector.** Explicitly, RE of type 1(2) for gene $i$ is $n_i(s)/n_i(s')$, where $n_i(s)$ is normalized expression for a sample $s$ of type 1(2) and $s'$ is a sample of empty vector. Type 1(2) corresponds to MT-2 cells transduced with TLL1-isoform 1 (iso-form 2). Genes were picked up if the RE of type 1 was larger than 11 and the RE of type 2 was smaller than 3. Also, Genes were picked up if the RE of type 1 was smaller than 1 and the RE of type 2 was larger than 5.

| Gene symbol | RE of type 1 | RE of type 2 |
|---|---|---|
| CCL3 | 14.53 | 2.94 |
| CCR6 | 11.70 | 2.85 |
| MIR155 | 11.30 | 2.44 |
| POSTN | 11.06 | 2.49 |
| HBG2 | 0.058 | 15.00 |
| HBB | 0.15 | 12.29 |
| HBA2 | 0.14 | 5.79 |
| HBA1 | 0.09 | 5.04 |

migration [39], microRNA-155 (MIR155) modulates Treg cell differentiation and its expression is up-regulated in HTLV-1 transformed cells [40–42], the chemokine CCL3 regulates myeloid differentiation [43], and POSTN has been reported to be involved in TGF-$\beta$ activation [44]. As for the four genes significantly up-regulated when TLL2-isoform 2 was transduced, all were globin genes. These findings reiterate the dependence of TLL's function on its isoforms in ATL cells.

## Discussion

In this paper, we statistically characterized the anomalous chromatin accessibility and gene expression of HTLV-1-infected cells and healthy CD4$^+$ T cells at the whole genome level.

Our analysis suggests that compared to healthy CD4$^+$T cells, ATL cells have the following properties: the chromatin accessibility increased near TSSs, a higher frequency of larger sample-to-sample fluctuations at the whole genome level, and a higher frequency of intermediate sample-to-sample fluctuations in gene coding regions. Consistently, histone-related genes were up-regulated. ATL cells had unique properties compared to CD4$^+$T cells from the viewpoint of the correlation between chromatin accessibility and gene expression. Further, whereas the immunophenotype determined by the systematically selected open chromatin regions was classified to be near CD4$^+$T cells for most samples, some samples were classified as myeloid cells.

Based on the above integrative analysis of chromatin accessibility and gene expression, we found that there were chromatin regions which were open in all the ATL cases but closed in all the analyzed samples of the 13 hematopoietic cell types derived from healthy donors. One of the genes overlapping with the chromatin regions that satisfy such conditions was *TLL1*, which was experimentally shown to have a large potential to regulate TGF-$\beta$. Indeed, the RNA-seq data of ATL samples imply that activation effect by TLL1 isoform 1 (NM_012464) on TGF-$\beta$ was dominant over repression effect by TLL1 isoform 2 (NM_001204760). Thus, we found that ATAC-seq data had information which leads us to find *TLL1* as a gene showing important gene expression. However, we have not yet investigated how TLL1 affects the disease from the aspect of symptom, which remains as an intriguing future study.

Contrary to ATL cases, the statistics of the chromatins in HAM cells resembled those of healthy CD4$^+$T cells, including sample-to-sample fluctuations. This observation implies that there was a certain sample-independent trend in the chromatin structure of HAM cases. Note that, due to the lack of proper RNA-seq data of HAM samples, we were unable to study the

gene expression pattern of TLL1 and its associated effect on TGF-$\beta$ in the HAM samples, which remain to be analyzed in future studies.

It should be noted that we were unable to analyze large number of samples due to the difficulty in obtaining samples for given clinical conditions. Thus, some quantities, such as frequencies shown in Figs 6, 7 and 8 were not estimated with statistical validity. To validate the hypothesis about the uniqueness of ATL cells across different scales from chromatin and transcription to immunophenotypes, more samples should be used in future studies.

Our finding about ATL samples might motivate us to consider a rather general relationship between increased chromatin accessibility and the onset of leukemia. Indeed, in a previous study of Acute Myeloid Leukemia (AML), it was reported that mutations in cohesin genes increase chromatin accessibility, which controls the activity of transcription factors leading to leukemogenesis [21]. It was also reported that HMGN1 amplification is also associated with increased chromatin accessibility; It conferred a transcriptional and chromatin phenotype associated with stem cells and leukemia [45]. It remains for future studies to check how the relationship between the increased chromatin accessibility and the onset of leukemia can be generalized beyond the above cases such as ATL and AML.

The previous studies reported that valemetostat, which is an epigenetic EZH1/2 inhibitor, has efficacy for ATL cases [46] and HAM cases [47]. Also, they elucidated the mechanism of resistance acquisition to valemetostat in patients with ATL [46]. Thus, epigenetic properties could play important role in understanding the nature of ATL cases. Our results obtained in this paper could shed light on epigenomic abnormalities in ATL from a different point of view. For example, Koya et al performed the transcriptome analysis to show that ATL samples are heterogeneous between patients, whereas non-malignant cells are similar between patients [48]. Our analysis shows that at the level of chromatin accessibility, ATL samples are heterogeneous as already discussed in the analysis with Fig 5.

As another example, Kogure et al. recently performed an analysis of large-scale whole genome sequencing for 150 ATL cases [49]. They investigated coding/noncoding mutations, structural variations, copy number alterations, and then identified 56 driver mutations in ATL cases. Using these driver mutational profiles, they found that ATL patients could be classified into two groups. These two groups differ in clinical findings and have significantly distinct prognoses. In this paper, we also found that ATL cases can be classified into, at least, two groups based on chromatin accessibility as already discussed in the analysis with Table 3. However, we have not yet been able to provide information that can lead to the identification of poor prognostic factors. To this point, we would like to contribute in the future by increasing the number of analyzed cases. Such studies could provide important information for predicting prognosis or selecting efficient drugs.

Let us discuss a potential universality in the classification result of ATAC-seq of ATL cases in the following. In order to compare ATL cases with another type of leukemia in terms of immunophenotypes, we performed the systematic clustering method [15] for ATAC-seq data (GSE85853) of cutaneous T-cell lymphoma (CTCL), which has been reported to have a clinical and histopathological phenotype similar to that of ATL [50]. As shown in Table 6, we found that the chromatin structures in some CTCL cases are closer to myeloid cells rather than CD4+T cells, though CTCL is conventionally classified as T-cell leukemia. This observation could shed light on reconsideration of the function of myeloid-like chromatin structures from a viewpoint of leukemia.

As an additional note, Fig 12b suggests that the chromatin regions around the TLL1 loci tend to be open also for the cases of HAM and HTLV-1 carrier (carrier). This observation indicates that open chromatin regions around TLL1 are not the only cause of leukemia onset.

**Table 6. Clustering results of ATAC-seq from 9 CTCL cases [50] in terms of immunophenotype using the method in [15].**

| SRR Number | first | second | third | Patient tag |
|---|---|---|---|---|
| 4044872 | Ery | Mono | CLP | Patient-11 on 0-th day |
| 4044873 | CD4$^+$T | CD8$^+$T | Ery | on 0-th day |
| 4044879 | Ery | Mono | CLP | Patient-39 on 0-th day |
| 4044880 | Ery | Mono | CLP | on 0-th day |
| 4044887 | Ery | Mono | CD4$^+$T | Patient-60 on 0-th day |
| 4044889 | Ery | Mono | CLP | Patient-61 on 0-th day |
| 4044890 | CD4$^+$T | CD8$^+$T | B | Patient-62 on 0-th day |
| 4044891 | CD4$^+$T | Ery | CD8$^+$T | on 0-th day |
| 4044892 | CD4$^+$T | CD8$^+$T | NK | Patient-1366 on 0-th day |

Rather, it suggests that open chromatin regions are potentially caused by the infection itself and related to the latent period or expansion of the virus.

This study is the first comprehensive analysis of open chromatin structures in ATL samples. We hope that the findings will deepen understanding of the ATL pathogenesis.

## Materials and methods

### Ethics statement

Experiments using clinical samples were conducted according to the principles expressed in the Declaration of Helsinki and approved by the Institutional Review Board of Kyoto University (permit numbers G310 and G204). ATL patients provided written informed consent for the collection of samples and subsequent analysis.

### Sequencing sample preparation

Peripheral blood mononuclear cells from ATL patients classified into either acute subtype or chronic subtype, HAM patients, and HTLV-1 carriers were thawed and washed with PBS containing 0.1% BSA. To discriminate dead cells, we used a LIVE/DEAD Fixable Dead Cell Stain Kit (Invitrogen). For cell surface staining, cells were stained with APC anti-human CD4 (clone: RPA-T4) (BioLegend) and anti-SynCAM (TSLC1/CADM1) mAb-FITC (MBL) antibodies for 30 minutes at 4 ℃ followed by washing with PBS. HTLV-1-infected cells (CADM1$^+$ and CD4$^+$) were purified by sorting with a FACS Aria (Beckman Coulter) to reach 98–99% purity. Data were analyzed using FlowJo software (Treestar). Soon after the sorting, 10000-50000 HTLV-1-infected cells were centrifuged and used for ATAC-seq. Total RNA was isolated from the remaining cells using an RNeasy Mini Kit (Qiagen). Library preparation and high-throughput sequencing were performed at Macrogen Inc. (Seoul, Korea). The diagnostic criteria and classification of the clinical subtypes of ATL were performed as previously described [51].

77 ATAC-seq datasets from 13 human primary blood cell types were obtained from the Gene Expression Omnibus (GEO) with accession number GSE74912 [31], and RNA-seq datasets of CD4$^+$T and CD8$^+$T cells from healthy donors were obtained from GSE74246 [31]. The ATAC-seq and RNA-seq data for ATL samples can be downloaded from DNA Data Bank of Japan (DBBJ). For the accession number, refer to Tables 3 and 4 respectively. In particular, the samples used for the analysis of RNA-seq are ATL8, ATL10, ATL21, and ATL24. Those clinical subtypes are all classified into acute subtype. Note that for ATAC-seq, DRR574511 is identical to DRR250714, DRR574513 is identical to DRR250710, DRR574524 is identical to DRR250711, and DRR574527 is identical to DRR250712. Further, for RNA-seq data,

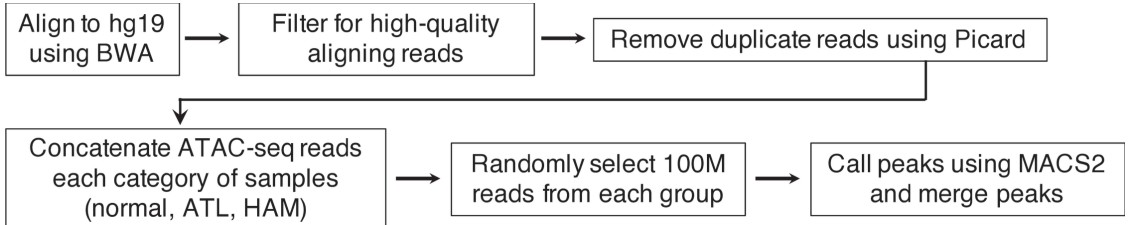

**Fig 14. Pipeline of the data processing for ATAC-seq.** The three steps at the bottom line correspond to the parts with Eqs (7), (8) and (9).

DRR573818 is identical to DRR250721, DRR573819 is identical to DRR250717, DRR573820 is identical to DRR250718, and DRR573821 is DRR250719. All the data starting with DRR2507 in the spelling were deposited for our previous study [15].

## Pre-processing of ATAC-seq

High-throughput sequencing provides a set of reads as output. ATAC-seq reads were aligned using BWA version 0.7.16a with default parameters. SAMtools was used to convert SAM files into compressed BAM files and sort the BAM files by chromosome coordinates. PICARD software (v1.119) (http://broadinstitute.github.io/picard/) was used to remove PCR duplicates using the `MarkDuplicates` options. Reads with mapping quality scores less than 30 were removed from the BAM files. For peak calling, MACS2 (v2.1.2) software was used with option `--nomodel --nolambda --keep-dup all -p 0.01`. This procedure is schematically described as a part of Fig 14. Note that the paired-end output of the sequence was used to reconstruct the fragments, where paired two reads correspond to both ends of a fragment.

Footprinting analysis was performed using HINT-ATAC as shown in Fig 10 [27]. Processed ATAC-seq BAM files were converted into TDF files by igvtools, which were visualized using Integrative Genomics Viewer (IGV) as shown in Fig 12 [52].

## Pre-processing of RNA-seq

RNA-seq data were aligned to human reference genome hg19 using STAR 2.6.1c with the `--quantMode GeneCounts` function [53]. RNA-seq data analysis was performed using edgeR, where the read counts of the RNA-seq data were normalized using TMM normalization [22] to be converted into pseudo read counts. As preliminary, the information about two classes such as ATL samples and CD4$^+$T samples is prepared, which is obtained by the command `factor`. Let $n_i^0(s)$ be the read counts of the RNA-seq data for each gene $i$ for a given cell sample $s \in \mathbb{S}$ and $N(s) := \sum_i n_i^0(s)$ be the total read counts over all genes, where $\mathbb{S}$ is the set of samples. Using the TMM normalization where $n_i^0(s)$ of gene $i$ for sample $s$, and the output of the command `factor` are the input data, one can obtain the normalization factor $r(s)$ for a given sample $s$ using the command `calcNormFactors` of edgeR. After acquiring $r(s)$, the pseudo read counts $n_i(s)$, which we call normalized expression, is calculated using the command `estimateCommonDisp` of edgeR. We used $n_i(s)$ as the starting point of the RNA-seq data analysis in the main text. Note that the normalization depends on $\mathbb{S}$.

An additional analysis was done to evaluate robustness in terms of normalization methods of RNA-seq data. We computed the geometric mean of $N(s)r(s)$ as $N_0 = \left(\prod_{s \in \mathbb{S}} r(s)N(s)\right)^{1/|\mathbb{S}|}$, where $|\mathbb{S}|$ is the number of elements of $\mathbb{S}$. We then checked whether the pseudo read counts $n_i(s)$ is close to a normalized read counts $n_i'(s) := n_i^0(s)\frac{N_0}{r(s)N(s)}$ for sample $s$. We found that the

maximum deviation between $n_i'(s)$ and $n_i(s)$ is smaller than 5 over all genes in the case of $\mathbb{S}$ equal to the set of 4 samples with CD4$^+$T and 4 samples with Mono. In this case, the effects from the differences between $n_i'(s)$ and $n_i(s)$ are quite small except for quantities related to genes with almost zero reads. Therefore, even if we use $n_i'(s)$ as the starting point of the analysis instead of the pseudo read counts $n_i(s)$, qualitatively the same conclusion as that with $n_i(s)$ is expected to be obtained.

The heatmap in Fig 6a was made using edgeR as follows; First, instead of using the command `estimateCommonDisp`, we used the command `estimateDisp` to get a normalized expression of RNA-seq data. Second, we assumed generalized linear models to fit the normalized expression, where we used the command `glmFit` to get the input of the command `glmLRT` with parameter `coef = 2`. Note that the command `glmFit` used the output of the command `factor` as an input. Thus, we chose the top 1000 reliable genes estimated by the generalizd linear models using the command `topTags`. Then, we calculated Counts per Million mapped reads (CPM) of the chosen genes by the command `cpm` with parameters of `log = TRUE` and `prior.count = 2`. Finally, we used the complete linkage method with euclidean distance between CPM of a pair of the chosen genes to obtain the hierarchical clustering.

Principal Component Analysis (PCA) in Fig 6b was done using the covariance matrix of $\log_{10}(n_i(s) + 1)$, where the first and second principal components were calculated using the `prcomp` command with option `scale = FALSE` in R (programming language).

In the case of transcript-level expression analysis of RNA-seq data, we used STAR 2.6.1c with the `--quantMode TranscriptomeSAM` option for mapping. Then, in order to estimate the isoform expression, we used RSEM version 1.3.1 with the `rsem-calculate-expression` function [54]. Then, the output data include the values of the transcripts per kilobase million (TPM), which we used for the comparison of expression level between TLL1 isoform 1 and isoform 2 in Table 4.

## Experimental setups

**Real-time PCR.** cDNA products were analyzed by real-time PCR using PowerUp SYBR Green Master Mix and StepOnePlus Real-Time PCR System (Applied Biosystems) according to the manufacturer's instructions. Primer sequences for the *GAPDH* gene have been described previously [55], and primer sequences for the *TLL1* gene were 5-TTGTTTTCTACGGGGAGCTATGG-3 and 5-ATATCGCCCCAAAATACAGCG-3. The relative quantification was calculated according to the method described in Applied Biosystems ABI prism 7700 SDS User Bulletin #2. Note that the ATL samples used for this experiment shown in Fig 12c are not listed in Table 3; those clinical subtypes are listed in Table 7.

**Luciferase assay.** HepG2 cells were cultured in DMEM supplemented with 10% FBS and antibiotics. The coding region of human TGF-$\beta$, whose length is 1173 bp, was generated by PCR amplification and subcloned into a pFUSE-hIgG1-Fc2 vector. HepG2 cells were plated on 12-well plates at $1 \times 10^5$ cells per well. After 24 hours, the cells were transfected with 50 ng/well of luciferase reporter plasmid (p3TP-Lux) [56], 5 ng/well of Renilla luciferase control vector (phRL-TK) together with 35 ng/well of TLL1 expression plasmid, and 35 ng/well of TGF-$\beta$-expressing plasmid or empty vector. Plasmids were transfected using TransIT-LT1 (Mirus) according to the manufacturer's instructions. After 48 hours, the cells were collected, and luciferase activities were measured using the Dual-Luciferase Reporter Assay Kit (Promega). Relative luciferase activity was calculated as the ratio of firefly to Renilla luciferase activity. Three independent experiments, each with triplicate transfections, were performed, and typical results are shown.

**Table 7. The clinical subtypes corresponding to the ATL samples shown in Fig 12c.**

| Sample labels | Clinical subtype |
| --- | --- |
| P1 | Lymphoma |
| P2 | Chronic |
| P3 | Chronic |
| P7 | Acute |
| P17 | Acute |
| P22 | Acute |
| P23 | Acute |
| P29 | Chronic |
| P41 | Acute |
| P48 | Acute |
| P49 | Acute |
| P54 | Acute |
| P59 | Chronic |
| P60 | Acute |
| P63 | Acute |
| P72 | Acute |
| P80 | Acute |
| P102 | Acute |
| P103 | Acute |

**Lentiviral vector construction and transfection of recombinant lentivirus.** The coding region of TLL1 isoform 2 was synthesized using gBlocks Gene Fragment (Integrated DNA Technolofies), which was used as the template for synthesizing TLL1 isoform 1 by PCR amplification. TLL1 isoform 1 (NM_012464) or isoform 2 (NM_001204760) fragments were subcloned into pCS2-EF-MCS (gift from H. Miyoshi, RIKEN Bioresource Center). An empty vector that expresses only hrGFP was used as the control for the lentiviral transduction. 293T cells at 80% confluence in a 10-cm dish were co-transfected with 10 $\mu$g lentivirus vector, 10 $\mu$g psPAX2, 5 $\mu$g pMD2.G, and PEI (Polyethylenimine). 48 hours after the transfection, supernatant containing the virus was collected and concentrated by ultracentrifugation. MT-2 cells were cultured in RPMI supplemented with 10% FBS and antibiotics. MT-2 cells were transfected with the lentivirus, and two weeks after the transduction, GFP-positive cells were purified by sorting with FACS Canto. RNA was isolated using the Qiagen RNeasy Mini Kit and then used for the RNA-seq analysis.

## Explicit definitions of computed quantities

We explicitly define quantities discussed in the main text. First, we assume that the set of reads from DNA of an ATAC-seq sample and the read counts for each gene from an RNA-seq sample are given. We also assume that a set of fragments for an ATAC-seq sample is given by using a pair of reads; both ends of a fragment correspond to a pair of two reads. The operational protocols are explained in Materials and Methods.

The positions of TSSs and coding regions of all genes were obtained from the human genome (hg19) as a set of intervals on the genome. Thus, a read from an ATAC-seq data is interval $[x_1, x_2]$ on the genome, which corresponds to a region including an edge of a fragment. A fragment has length $\ell$ and location $x$ as the mid-point of the two edges on the genome [15].

The reads from an RNA-seq sample provide the normalized expression $n_i(s)$ for each gene $i$, where $s \in \mathbb{S}$ is the sample index. We denote by $\mathbb{S}_v$ the set of all analyzed samples with type $v$.

**The normalized number of reads in Fig 3a.** Let us consider the number $\rho_s(z)$ of ATAC-seq reads from a sample $s \in \mathbb{S}$ located on position $z$ from the nearest TSS. Then, we take the sample average among type $v$ as

$$\overline{\rho}_v(z) := \frac{1}{|\mathbb{S}_v|} \sum_{s \in \mathbb{S}_v} \rho_s(z), \tag{3}$$

where $v \in \{\text{CD4}^+\text{T, HAM, ATL}\}$. In Fig 3a, we plot the normalized quantity $\tilde{\rho}_v(z)$ obtained after dividing $\overline{\rho}_v(z)$ by the value at the TSS ($z = 0$) such that

$$\tilde{\rho}_v(z) := \frac{\overline{\rho}_v(z)}{\overline{\rho}_v(0)}. \tag{4}$$

**The averaged number of fragments in Fig 3b.** Let us consider the number of fragments $\phi_s(z, \ell)$ from sample $s \in \mathbb{S}$ satisfying the following two conditions: (i) their centers are located at $z$, and (ii) they have length $\ell$. $\overline{\phi}_v(z, \ell)$ describes the sample average among type $v$ as

$$\overline{\phi}_v(z, \ell) := \frac{1}{|\mathbb{S}_v|} \sum_{s \in \mathbb{S}_v} \phi_s(z, \ell), \tag{5}$$

where $v \in \{\text{CD4}^+\text{T, HAM, ATL}\}$. In Fig 3b, we plot the histogram $F_v^{\Delta, \xi}(z, \ell)$ for bin width $\Delta$ and $\xi$ for $z$ and $\ell$, respectively, as

$$F_v^{\Delta, \xi}(z, \ell) := \sum_{z - \Delta/2 \leq z' < z + \Delta/2} \sum_{\ell \leq \ell' < \ell + \xi} \overline{\phi}_v(z', \ell'). \tag{6}$$

**The reference set of peaks.** To analyze open chromatin regions, we used MACS2 with the input of reads from ATAC-seq data. Concretely, we used MACS2 with the option `--nomodel --nolambda --keep-dup all -p 0.01`, which corresponds to $p_G = 10^{-2}$ [15]. This algorithm outputs the collection of peaks $\hat{g}_s$ for a given sample $s$ as candidates of open chromatin regions, which can be described as

$$\hat{g}_s := ((\gamma_k, \alpha_k, \beta_k), p_k)_{k \geq 1}, \tag{7}$$

where $\gamma_k$ is the chromosome number, $\alpha_k$ is the starting point, and $\beta_k$ is the ending point in terms of genome position with $p_k$ as the $p$-value of the $k$-th peak. As in [15], $p_k \leq p_{k'}$ for $k < k'$. In particular, we consider the set of the top $M$ peaks and denote it by

$$\hat{g}_s^M := \begin{cases} ((\gamma_k, \alpha_k, \beta_k), p_k)_{k=1}^M & (\text{if } |\hat{g}_s| \geq M), \\ \hat{g}_s & (\text{otherwise}). \end{cases} \tag{8}$$

Next, we concatenate the data of all reads from all ATAC-seq samples with cell type $v \in \{\text{CD4}^+\text{T, HAM, ATL}\}$. Then, we randomly extract 100 million reads from the concatenated data for type $v$ as the input for the MACS2 algorithm to obtain the collection of peaks,

$$\hat{g}_v := ((\gamma_k, \alpha_k, \beta_k), p_k)_{k \geq 1}. \tag{9}$$

Using a coalescing process of $\hat{g}_{\text{ATL}}, \hat{g}_{\text{HAM}}, \hat{g}_{\text{CD4}^+\text{T}}$, we construct a new reference set of peaks $\hat{g}_0$ as follows. Operationally, the coalescing of two peaks is done as follows. If two peaks ($\gamma$, $\alpha$,

$\beta$) and $(\gamma', \alpha', \beta')$ for $\gamma = \gamma'$ satisfy $\alpha' \leq \alpha \leq \beta'$, the two peaks become one peak as $(\gamma, \alpha', \max\{\beta, \beta'\})$. This operation is repeated for a newly obtained set of peaks until no more coalescing processes occur. In this subsection, for simplicity, chromosome number is often ignored because the following discussions can be straightforwardly reconstructed in a similar manner even with chromosome number.

**The width of overlapped peaks in Figs 4 and 5.** To quantify the similarity between two collection of peaks, $\hat{g}_0$ and $\hat{g}_s^M$, first, we fix a set of peaks $\tilde{g}$ as any of (i) the set $\tilde{g}_c$ of all peaks in $\hat{g}_0$ overlapping gene coding regions, (ii) the set $\tilde{g}_{nc}$ of all peaks in $\hat{g}_0$ corresponding to non-coding regions, and (iii) the union $\tilde{g}_c \cup \tilde{g}_{nc}$ of two sets $\tilde{g}_c, \tilde{g}_{nc}$. Note that for a given peak, its center was calculated by the command of `annotatePeaks.pl` in the HOMER algorithm and used to judge whether the peak joins $\tilde{g}_c$ or $\tilde{g}_{nc}$ [57].

Then, for a region $(\alpha, \beta)$, we define $O((\alpha, \beta), \hat{g}_s^M)$ as the number of base pairs in a peak of $\hat{g}_s^M$. Namely, $O((\alpha_\kappa[\tilde{g}], \beta_\kappa[\tilde{g}]), \hat{g}_s^M)$ is that for the $\kappa$-th peak $(\alpha_\kappa[\tilde{g}], \beta_\kappa[\tilde{g}])$ of $\tilde{g} \subset \hat{g}_0$. Then, we compute the average and variance of $O((\alpha_\kappa[\tilde{g}], \beta_\kappa[\tilde{g}]), \hat{g}_s^M)$ as follows:

$$\overline{O}_\kappa(\tilde{g}, \mathbb{S}) := \frac{1}{|\mathbb{S}|} \sum_{s \in \mathbb{S}} O((\alpha_\kappa[\tilde{g}], \beta_\kappa[\tilde{g}]), \hat{g}_s^M), \tag{10}$$

$$V_\kappa(\tilde{g}, \mathbb{S}) := \frac{1}{|\mathbb{S}|} \sum_{s \in \mathbb{S}} \left( O((\alpha_\kappa[\tilde{g}], \beta_\kappa[\tilde{g}]), \hat{g}_s^M) - \overline{O}_\kappa(\tilde{g}, \mathbb{S}) \right)^2. \tag{11}$$

We set $M = 64000$ as the provisionally optimal number for immunophenotype classification [15]. Note that letting $(\alpha_{\text{TREC}}, \beta_{\text{TREC}})$ and $(\alpha_{\text{TRA}}, \beta_{\text{TRA}})$ be the region of TREC and TRA genes respectively, $O((\alpha_{\text{TREC}}, \beta_{\text{TREC}}), \hat{g}_s^M)$ is the total width of the peaks overlapping with TREC region mentioned in the caption of Fig 9. Explicitly, $\alpha_{\text{TREC}} = 22, 855, 260, \beta_{\text{TREC}} = 22, 944, 268$ and $\alpha_{\text{TRA}} = 22, 090, 057, \beta_{\text{TRA}} = 23, 021, 075$. The chromosome number of TRA region is $\gamma = 14$, and TRA region includes TREC region.

The following functions describe the frequency of the average and variance of $O((\alpha_\kappa[\tilde{g}], \beta_\kappa[\tilde{g}]), \hat{g}_s^M)$:

$$\rho_{\tilde{g}, \mathbb{S}}^{(1)}(O) := \sum_{\substack{\kappa \\ (\alpha_\kappa, \beta_\kappa) \in \tilde{g}}} \delta(O, \overline{O}_\kappa(\tilde{g}, \mathbb{S})), \tag{12}$$

$$\rho_{\tilde{g}, \mathbb{S}}^{(2)}(V) := \sum_{\substack{\kappa \\ (\alpha_\kappa, \beta_\kappa) \in \tilde{g}}} \delta(V, V_\kappa(\tilde{g}, \mathbb{S})), \tag{13}$$

where $\kappa$ runs the indices of peaks such that $(\alpha_\kappa, \beta_\kappa) \in \tilde{g}$, and $\delta(a, b) = 1$ for $a = b$, otherwise 0.

Lastly, in Figs 4 and 5, the histograms for bin width $\Delta$ for $O$ is defined as

$$F_{\tilde{g}, \mathbb{S}}^{(1)}(O; \Delta) := \sum_{O - \Delta/2 \leq O' < O + \Delta/2} \rho_{\tilde{g}, \mathbb{S}}^{(1)}(O'), \tag{14}$$

$$F_{\tilde{g}, \mathbb{S}}^{(2)}(V; \Delta) := \sum_{V - \Delta/2 \leq V' < V + \Delta/2} \rho_{\tilde{g}, \mathbb{S}}^{(2)}(V'). \tag{15}$$

**Log-fold change of gene expression and chromatin accessibility in Figs 7 and 8.** We consider the set of cell types $\mathbb{T}$ as

$$\mathbb{T} = \{\text{HSC}, \text{CD4}^+\text{T}, \text{CD8}^+\text{T}, \text{Mono}, \text{ATL}\}. \tag{16}$$

For a gene $i$, we compute the fold change $\text{FC}_i^A(t_1, t_2)$ of ATAC-seq data in gene $i$ between types $t_1, t_2 \in \mathbb{T}$ as

$$\text{FC}_i^A(t_1, t_2) := \frac{\overline{A}_i(t_1)}{\overline{A}_i(t_2)}, \tag{17}$$

where $\overline{A}_i(t)$ is the average, over all samples with type $t$, width of the peaks of ATAC-seq data overlapping with the gene $i$. Similarly, for a gene $i$, we compute the fold change $\text{FC}_i^R(t_1, t_2)$ of RNA-seq data in gene $i$ between types $t_1, t_2 \in \mathbb{T}$ as

$$\text{FC}_i^B(t_1, t_2) := \frac{\overline{B}_i(t_1)}{\overline{B}_i(t_2)}, \tag{18}$$

where $\overline{B}_i(t)$ is the average, over all samples with type $t$, of normalized expression $n_i$ of the RNA-seq data for gene $i$. Note that the normalization, which is TMM normalization by edgeR, is done together for all the samples of a given pair of type $t_1$ and type $t_2$.

Here, we consider only the genes, for which both $\log_2 \text{FC}_i^A(t_1, t_2)$ and $\log_2 \text{FC}_i^R(t_1, t_2)$ are well-defined; in other words, we only consider the genes $i$ satisfying all the four conditions (i) $\overline{A}_i(t_1) \neq 0$, (ii) $\overline{A}_i(t_2) \neq 0$, (iii) $\overline{B}_i(t_1) \neq 0$, and (iv) $\overline{B}_i(t_2) \neq 0$. For example, the condition (i) means that there is, at least, one peak among the top M peaks of, at least, one sample with type $t$ in $\mathbb{S}_t$, which intersects with the coding regions of the gene $i$; the condition (iii) means that there is, at least, one read among all the sample with type $t$ in $\mathbb{S}_t$, which intersects with the coding regions of the gene $i$. We define a set $\mathbb{G}_{t_1, t_2}^M$ as the set of all the genes satisfying that both $\log_2 \text{FC}_i^A(t_1, t_2)$ and $\log_2 \text{FC}_i^B(t_1, t_2)$ for gene $i$ are well-defined.

In Fig 7, we focus on the following function quantifying the frequency of the log-fold change

$$\rho_{t_1, t_2}^M(P_A, P_B) := \sum_{i \in \mathbb{G}_{t_1, t_2}^M} \delta(P_A, \log_2 \text{FC}_i^A(t_1, t_2))\delta(P_B, \log_2 \text{FC}_i^B(t_1, t_2)). \tag{19}$$

Further, we transform the variables from $(P_A, P_B)$ to $(R, \theta)$ by using $P_A = R \cos\theta$ and $P_B = R \sin\theta$. Then, we define a distribution function

$$\eta_{t_1, t_2}^M(R, \theta) := \rho_{t_1, t_2}^M(R \cos\theta, R \sin\theta)/R. \tag{20}$$

In Figs 7 and 8, we plot the histogram with distribution function $\eta_{t_1, t_2}^M(R, \theta)$ for bin width of $R$ and $\theta$ equal to 0.1 and 6, respectively.

**$\Delta\Delta C_t$ method in Fig 12c.** To obtain the threshold cycle for real-time PCR of an mRNA sample (see "Real-time PCR" for details), for sample $s \in \mathbb{S}$, let $C_s^{TLL1}$ denote the threshold cycle for gene *TLL1* and $C_s^{GAPDH}$ does for the gene *GADPH*. Then, we define the difference $\Delta C_s := C_s^{TLL1} - C_s^{GAPDH}$ and consider the normalized difference as

$$\Delta C_s^0 := \Delta C_s - \min_{s \in \mathbb{S}} \Delta C_s. \tag{21}$$

## Acknowledgments

We thank P. Karagiannis for proofreading the manuscript and many valuable comments.

## Author Contributions

**Conceptualization:** Azusa Tanaka.

**Data curation:** Azusa Tanaka.

**Formal analysis:** Azusa Tanaka, Yasuhiro Ishitsuka, Hiroki Ohta.

**Funding acquisition:** Azusa Tanaka, Hiromitsu Tanaka, Jun-ichirou Yasunaga, Masao Matsuoka.

**Investigation:** Azusa Tanaka, Chiho Onishi, Hiromitsu Tanaka, Jun-ichirou Yasunaga, Masao Matsuoka.

**Methodology:** Azusa Tanaka, Yasuhiro Ishitsuka, Hiroki Ohta, Jun-ichirou Yasunaga.

**Project administration:** Masao Matsuoka.

**Resources:** Norihiro Takenouchi, Masanori Nakagawa, Ki-Ryang Koh, Akihiro Fujimoto, Jun-ichirou Yasunaga, Masao Matsuoka.

**Supervision:** Akihiro Fujimoto, Jun-ichirou Yasunaga, Masao Matsuoka.

**Validation:** Azusa Tanaka, Yasuhiro Ishitsuka, Hiroki Ohta.

**Visualization:** Azusa Tanaka, Yasuhiro Ishitsuka, Hiroki Ohta.

**Writing – original draft:** Azusa Tanaka, Yasuhiro Ishitsuka, Hiroki Ohta.

**Writing – review & editing:** Azusa Tanaka, Yasuhiro Ishitsuka, Hiroki Ohta, Masao Matsuoka.

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
