## [Decision Letter · Decision Letter 0]

8 May 2024

Dear Dr. Tanaka,

Thank you very much for submitting your manuscript "Integrative analysis of ATAC-seq and RNA-seq for cells infected by human T-cell leukemia virus type 1" for consideration at PLOS Computational Biology.

As with all papers reviewed by the journal, your manuscript was reviewed by members of the editorial board and by several independent reviewers. In light of the reviews (below this email), we would like to invite the resubmission of a significantly-revised version that takes into account the reviewers' comments.

We cannot make any decision about publication until we have seen the revised manuscript and your response to the reviewers' comments. Your revised manuscript is also likely to be sent to reviewers for further evaluation.

Sincerely,

Ilya Ioshikhes

Section Editor

PLOS Computational Biology

Ilya Ioshikhes

Section Editor

PLOS Computational Biology

Reviewer's Responses to Questions

**Comments to the Authors: **

Reviewer #1: Tanaka and coleagues studied about the relationship between chromattin accesibility and pathogenesis of ATL and HAM, especially ATL. It is interesting work but first roud reviewers pointed out many concerns.The authors should reply carefully to the comments. 

1) In comment 1-2, the refree pointed out the lack of biological consideratiions and new discoveries about ATL /HAM, but the authors replied by the comments about CTCL. They should reply about the relevance concerning ATL/HAM. Moreover in the discussion about CTCL, the relationship between romidepsin treatment reponsiveness and pre-treatment chromatin structure is unclear.

2) In comment 1-3, the refree commented about significnace of chromatin conformational chages for the disease, but your reply did not refer to the point. FUrther 3d structural changes were not outo of scope in this comment.

3)Incomment 1-4, the authors should reply about overall expression pattern such as heamap. Revision 1-4 did not reply to the refree's commnet.

4) The expressoin level of TLL1 isoform 1 is about 10.3 times higher than that of TLL isoform 2. The authors should show the data.

5) As pointed out in comment 1-5, chromatin conformational changes in TLL1 have also been observed in Fig 10B. The refree asked the authors to consider whether the effects of TGF are also likely to be affected in HAM, but there were no consideration in the authors' revision.

6) As pointed out in comment1-6, I also think that footprint and other data should be moved to the results section and discussed in the discussion sectiion.

7) It is not clear that remidepsin treatment responsiveness has relationship with pre-treatment chromatin structure from the data shown in the manuscript.

Reviewer #2: This study investigates the interplay between chromatin accessibility and transcription in HTLV-1-infected cells at a genome-wide scale, utilizing ATAC-seq and RNA-seq data. By employing a systematic clustering algorithm, the authors compared HTLV-1 infected CD4+ cells from ATL and HAM cases with healthy CD4+ T cells. They observed abnormal properties in the open chromatin regions of CD4+ T cells derived from ATL cases compared to healthy counterparts, along with distinct gene expression patterns between the two groups based on immunophenotype. Furthermore, the analysis revealed a relationship between chromatin accessibility and immunophenotype, indicating a potential resemblance of some ATL cases to various myeloid cell types. Notably, the study identified TLL1 as one of the genes with aberrant expression in ATL cases, with subsequent experiments demonstrating isoform-dependent regulation of TGF-β maturation by TLL1 isoforms, which is relevant to cancer progression. This study is well-designed and has uncovered novel findings, shedding light on previously unexplored aspects of ATL. I have a few minor comments.

1)HTLV-1 infected CD4+ cells are thought to be consisted mainly of leukemic cells in ATL. If so, it would be better to clearly state this.

2)It would be better to make discussions based on more recent studies using human ATL and HAM samples. For example, there have been several papers investigating i) epigenetic status using ChIP-seq and methylation microarray, ii) non-coding genome using whole-genome sequencing, and iii) immunophenotypes and heterogeneity (or fluctuations) within and among patients using single-cell analysis in ATL.

3)Gene name should be italicized.

**Have the authors made all data and (if applicable) computational code underlying the findings in their manuscript fully available?**

Reviewer #1: Yes

Reviewer #2: **No: **If the accession number for sequence data is issued, please provide it. Please provide the computational code used in this study.

PLOS authors have the option to publish the peer review history of their article (what does this mean?). If published, this will include your full peer review and any attached files.

Reviewer #1: No

Reviewer #2: **Yes: **Keisuke Kataoka
---

## [Decision Letter · Decision Letter 1]

18 Sep 2024

Dear Dr. Tanaka,

Thank you very much for submitting your manuscript "Integrative analysis of ATAC-seq and RNA-seq for cells infected by human T-cell leukemia virus type 1" for consideration at PLOS Computational Biology. As with all papers reviewed by the journal, your manuscript was reviewed by members of the editorial board and by several independent reviewers. The reviewers appreciated the attention to an important topic. Based on the reviews, we are likely to accept this manuscript for publication, providing that you modify the manuscript according to the review recommendations.

Sincerely,

Ilya Ioshikhes

Section Editor

PLOS Computational Biology

Ilya Ioshikhes

Section Editor

PLOS Computational Biology

Reviewer's Responses to Questions

**Comments to the Authors: **

Reviewer #1: The mansucript is well-revised and now worth to be publieshed.

I have some minor comments and question. Please consider moinor modification.

1) This study analyzed the peripheral blood sample of ATL and HAM patients, not asymptomatic carriers. So, the results of this study is not necessarily reflect the change caused by HTLV-1 infection alone. The authors should be careful to this point. Also, to analyze aggresive ATL cells, CADM1+/CD7- fraction should be analyzed. CD4+/CADM1+ fraction of the peripheral blood of ATL patinets include true aggressive tumor cells and those which are HTLV-1 ifected but not fully transformed.

2)In seciton F. I cannot understand why it is considered that HTLV-1 infected cells have already differiated into T cells before infection. Is the possibility that HTLV-1 infected cell differetiate into T cell and TCR rearrangement occur ater infection is excluded?

Reviewer #2: The authors successfully addressed the reviewer's concerns. Please make sure to make all data and computational code underlying the findings fully available.

**Have the authors made all data and (if applicable) computational code underlying the findings in their manuscript fully available?**

Reviewer #1: Yes

Reviewer #2: **No: **The authors wrote that all sequence data has been uploaded to DDBJ and is waiting for an accession number to be issued. In addition, there are no statement regarding code availability. Please make sure to make all data and computational code underlying the findings fully available.

PLOS authors have the option to publish the peer review history of their article (what does this mean?). If published, this will include your full peer review and any attached files.

Reviewer #1: No

Reviewer #2: **Yes: **Keisuke Kataoka

Figure Files:

Data Requirements:

Reproducibility:

References:

---

## [Decision Letter · Decision Letter 2]

28 Oct 2024

PCOMPBIOL-D-24-00264R2Integrative analysis of ATAC-seq and RNA-seq for cells infected by human T-cell leukemia virus type 1PLOS Computational Biology Dear Dr. Tanaka, Thank you for submitting your manuscript to PLOS Computational Biology. After careful consideration, we feel that it has merit but does not fully meet PLOS Computational Biology's publication criteria as it currently stands. Therefore, we invite you to submit a revised version of the manuscript that addresses the points raised during the review process. Please submit your revised manuscript within 30 days Dec 28 2024 11:59PM. If you will need more time than this to complete your revisions, please reply to this message or contact the journal office at ploscompbiol@plos.org. Please include the following items when submitting your revised manuscript:*
A rebuttal letter that responds to each point raised by the editor and reviewer(s). You should upload this letter as a separate file labeled 'Response to Reviewers'. This file does not need to include responses to formatting updates and technical items listed in the 'Journal Requirements' section below.*
A marked-up copy of your manuscript that highlights changes made to the original version. You should upload this as a separate file labeled 'Revised Manuscript with Track Changes'.*
An unmarked version of your revised paper without tracked changes. You should upload this as a separate file labeled 'Manuscript'. If you would like to make changes to your financial disclosure, competing interests statement, or data availability statement, please make these updates within the submission form at the time of resubmission. Guidelines for resubmitting your figure files are available below the reviewer comments at the end of this letter. We look forward to receiving your revised manuscript. Kind regards, Ilya IoshikhesSection EditorPLOS Computational Biology Ilya IoshikhesSection EditorPLOS Computational Biology

Feilim Mac Gabhann

Editor-in-Chief

PLOS Computational Biology

Jason Papin

Editor-in-Chief

PLOS Computational Biology

 **Journal Requirements:** **Additional Editor Comments (if provided):** 

**Reviewers' comments:** Reviewer's Responses to Questions

**Comments to the Authors: **

Reviewer #1: The authors has now properly responded to my comments and the mansucript is worth to be published. I'd like to confirm one point whether the ATL patients analysed in this study are aggressive ATL patients. If so, the authors should describe acute type ATLpatients in stead of ATL patients in MATERIALS AND METHODS part. In acute type ATL patients, most of the cells sorted by Flow cytometry as CD4+/CADM1+ are tumor cells.

Reviewer #2: The authors successfully addressed my concerns.

**Have the authors made all data and (if applicable) computational code underlying the findings in their manuscript fully available?**

Reviewer #1: Yes

Reviewer #2: Yes

PLOS authors have the option to publish the peer review history of their article (what does this mean?). If published, this will include your full peer review and any attached files.

Reviewer #1: No

Reviewer #2: **Yes: **Keisuke Kataoka

---

## [Editor Report · Decision Letter 3]

2 Dec 2024

Dear Dr. Tanaka,

We are pleased to inform you that your manuscript 'Integrative analysis of ATAC-seq and RNA-seq for cells infected by human T-cell leukemia virus type 1' has been provisionally accepted for publication in PLOS Computational Biology.

Best regards,

Ilya Ioshikhes

Section Editor

PLOS Computational Biology

Ilya Ioshikhes

Section Editor

PLOS Computational Biology

Feilim Mac Gabhann

Editor-in-Chief

PLOS Computational Biology

Jason Papin

Editor-in-Chief

PLOS Computational Biology

---

## [Editor Report · Acceptance letter]

20 Dec 2024

PCOMPBIOL-D-24-00264R3 

Integrative analysis of ATAC-seq and RNA-seq for cells infected by human T-cell leukemia virus type 1

Dear Dr Tanaka,

I am pleased to inform you that your manuscript has been formally accepted for publication in PLOS Computational Biology. Your manuscript is now with our production department and you will be notified of the publication date in due course.

With kind regards,

Lilla Horvath
